# KLF4-PFKFB3-driven glycolysis is essential for phenotypic switching of vascular smooth muscle cells

Xinhua Zhang [iD] [1✉], Bin Zheng[1], Lingdan Zhao[1], Jiayi Shen[1], Zhan Yang[1,2], Yu Zhang[1], Ruirui Fan[1], Manli Zhang[1,3], Dong Ma[1], Lemin Zheng[4], Mingming Zhao[4], Huirong Liu[5] & Jinkun Wen[1✉]

Vascular smooth muscle cells (VSMCs) within atherosclerotic lesions undergo a phenotypic switching in a KLF4-dependent manner. Glycolysis plays important roles in transdifferentiation of somatic cells, however, it is unclear whether and how KLF4 mediates the link between glycolytic switch and VSMCs phenotypic transitions. Here, we show that KLF4 upregulation accompanies VSMCs phenotypic switching in atherosclerotic lesions. KLF4 enhances the metabolic switch to glycolysis through increasing PFKFB3 expression. Inhibiting glycolysis suppresses KLF4-induced VSMCs phenotypic switching, demonstrating that glycolytic shift is required for VSMCs phenotypic switching. Mechanistically, KLF4 upregulates expression of circCTDP1 and eEF1A2, both of which cooperatively promote PFKFB3 expression. TMAO induces glycolytic shift and VSMCs phenotypic switching by upregulating KLF4. Our study indicates that KLF4 mediates the link between glycolytic switch and VSMCs phenotypic transitions, suggesting that a previously unrecognized KLF4-eEF1A2/circCTDP1-PFKFB3 axis plays crucial roles in VSMCs phenotypic switching.

[1] Department of Biochemistry and Molecular Biology, the Key Laboratory of Neural and Vascular Biology, Ministry of Education of China, Hebei Medical University, Shijiazhuang, China. [2] Department of Urology, The Second Hospital of Hebei Medical University, Shijiazhuang, China. [3] Department of Emergency Medicine, The Second Hospital of Hebei Medical University, Shijiazhuang, China. [4] The Institute of Cardiovascular Sciences and Institute of Systems Biomedicine, School of Basic Medical Sciences, and Key Laboratory of Molecular Cardiovascular Sciences of Ministry of Education, Health Science Center, Peking University, Beijing, China. [5] Department of Physiology and Pathophysiology, School of Basic Medical Sciences, Capital Medical University, Beijing, China. ✉email: 18200814@hebmu.edu.cn; wjk@hebmu.edu.cn

Atherosclerosis is a chronic inflammatory disease of the vessel wall. Myocardial infarction, stroke and sudden cardiac death caused by atherosclerosis are the leading cause of death in the world[1,2]. Upon atherogenic stimuli, vascular smooth muscle cells (VSMCs), monocyte-derived macrophages (Mϕs) and endothelial cells (ECs) of the arterial wall actively participate in the development and progression of atherosclerosis[1,3]. It is well established that VSMCs in atherosclerotic lesions undergo phenotypic switching in response to atherogenic stimuli and acquire the capacity to secrete growth factors, proinflammatory cytokines and extracellular matrix (ECM) proteins that mediate various cellular responses[2,4]. In atherosclerotic lesions, some VSMCs can convert to Mϕ- and mesenchymal stem cell (MSC)-like cells that may further contribute to plaque formation[5]. VSMCs in human atherosclerotic lesions also express plasmacytoid dendritic cell (pDC) marker CD123 (interleukin-3 receptor α chain, IL3Rα) and CD123-positive pDCs have been detected in atherosclerotic plaques by electron microscopy and immunohistochemistry[6,7]. In addition, using flow cytometric analyses of freshly dissociated cells from the mouse aortic root, Owens et al. demonstrated that VSMCs can express dendritic cell marker ITGAX (CD11c)[5]. Thus, it is likely that VSMCs also have the potential to switch to DCs. KLF4, a critical factor in embryonic stem cell pluripotency maintenance and somatic cell reprogramming[8,9], is well known to be required for phenotypic switching of VSMCs in response to pro-atherosclerotic stimuli, such as platelet-derived growth factor-BB (PDGF-BB)[10], oxidized phospholipids[11], or IL1β[12]. Over the past 10 years, molecular mechanism mediating VSMC phenotypic switching by KLF4 has been extensively studied, and it has been demonstrated that KLF4 suppresses expression of VSMC marker genes by direct binding to the G/C repressor element present in most VSMC marker gene promoters[13], thereby modulating VSMC phenotypic transitions. Moreover, it is also found that the coordinated suppression of VSMC marker gene expression by KLF4 is accompanied by epigenetic chromatin modifications at VSMC marker gene loci[14,15]. While it is well established that KLF4 acts as a molecular fate switch promoting the VSMC-to-macrophage transdifferentiation, little is known about the role KLF4 plays in regulating VSMC transitions to other plaque cell types, such as DCs.

Increasing evidence suggests that cell reprogramming[16,17] and transdifferentiation[18,19] are accompanied by metabolic remodeling. For example, switching of ECs from a quiescent state to a highly migratory and proliferative state is driven by increased glycolysis[20]. Altered glucose metabolism and the overactivation of glucose-6-phosphate dehydrogenase (G6PD) switch pulmonary artery smooth muscle cells from the contractile to synthetic phenotype[21]. Moreover, metabolic switch from oxidative to glycolytic occurs during iPSC formation[22], and inhibition of glycolysis reduces the reprogramming efficiency while stimulation of glycolysis enhances iPSC generation[16]. However, there is little known about the role of KLF4 in metabolic reprogramming during VSMC phenotypic switching induced by atherogenic stimuli. Considering the crucial role of KLF4 in multiple biological processes, we hypothesized that VSMC phenotypic switching triggered by KLF4 could be driven by KLF4-mediated metabolic remodeling.

In glycolytic flux, the conversion of fructose-6-phosphate (F6P) to fructose-1,6-bisphosphate (F1,6P$_2$) is the first irreversible reaction catalyzed by 6-phosphofructo-1 kinase (PFK-1). PFK-1 is activated by its allosteric activator, fructose-2,6-bisphosphate (F2,6P$_2$), which is the most potent stimulator of glycolysis and is controlled by phosphofructokinase-2/fructose-2,6-bisphosphatase (PFKFB) enzymes[23]. Of all PFKFB isoenzymes, PFKFB3 has a much higher kinase/phosphatase activity ratio (700-fold), thus favoring the production of intracellular F2,6P$_2$ and glycolytic flux[24]. In this study, we showed that KLF4 can enhance glycolytic shift by upregulating PFKFB3 expression. Regarding the mechanism underlying PFKFB3 upregulation by KLF4, we found that KLF4 promotes the expression of circCTDP1 and the eukaryotic elongation factor eEF1A2, both of which cooperatively induce PFKFB3 expression by their interaction at the translation level, thereby shifting glucose metabolism toward glycolysis. We also reveal that trimethylamine-N-oxide (TMAO), a gut-microbiota-dependent metabolite, which is shown to enhance atherosclerosis in animal models and is associated with cardiovascular risks in clinical studies[25], induces the glycolytic shift and phenotypic switching of VSMCs in vitro and in vivo by upregulating KLF4 expression. Our findings indicate that a previously unrecognized KLF4-eEF1A2/circCTDP1-PFKFB3 regulatory pathway is involved in the phenotypic switching of VSMCs induced by pro-atherosclerotic factors including TMAO.

## Results

### KLF4 plays a critical role in triggering pDC marker expression in VSMCs.

First, we showed that CD123 and KLF4 were abundantly expressed in human atherosclerotic lesions (Fig. 1a, b). Then, using a multiplex tyramide signal amplification (TSA) staining, we observed 19% and 12% of SM α-actin positive (SMA-α$^+$) cells that expressed CD123 and CD68, respectively. Importantly, all CD123$^+$SMA-α$^+$ and CD68$^+$SMA-α$^+$ cells were KLF4 positive (Fig. 1c, d and Supplementary Fig. 1). Next, we performed an in situ hybridization proximity ligation assay (ISH-PLA) which permits identification of even phenotypically modulated VSMCs within fixed tissues based on detection of H3K4dime of the *Myh11* promoter (PLA$^+$), a SMC-specific epigenetic signature that persists in cells that have no detectable expression of SMC markers[5]. As a result, we observed a PLA signal in CD123$^+$ and LY6D$^+$ (another pDC marker) cells (Supplementary Fig. 2a–e). We next infected VSMCs with a recombinant adenovirus expressing KLF4 (pAd-GFP-KLF4) and performed mRNA microarray analyses to compare the gene expression profiling between KLF4-overexpressing VSMCs and control cells. The experiments showed that *CD123* and other DC markers such as *CXCR4, LILRB2, LILRB3, HMHA1, CD52, LAG3, CD83, CD24, TNFSF9, LY6D, TNFAIP2,* and *ZBTB46* were largely induced by KLF4 overexpression in VSMCs (Fig. 1e). These findings were further validated by quantitative RT-PCR (Fig. 1f). The specificity of KLF4-induced pDC marker expression was confirmed by the fact that KLF5 overexpression did not affect these markers (Fig. 1f). In addition, flow cytometry also confirmed the unique expression of CD123 in KLF4-overexpressing VSMCs (Fig. 1g). Furthermore, confocal immunofluorescent staining revealed that CD123 expression was exclusively observed in KLF4-overexpressing VSMCs that histologically resembled plasma pDC (Fig. 1h). These findings suggest that VSMCs within the plaques may switch to a pDC-like phenotype and that KLF4 upregulation is correlated with this phenotypic change. To further test the ability of VSMC-derived pDC-like cells to produce interferon alpha (IFN-α), which is an important function of maturated pDCs, we treated KLF4-overexpressing VSMCs with CpG oligodeoxynucleotides (ODN), an inducer of production of IFN-α, and measured the IFN-α levels in medium. As a result, IFN-α levels were not affected by KLF4 overexpression even in the presence of CpG-ODN (Supplementary Fig. 2e), indicating that VSMC-derived pDC-like cells may not have the functional properties of pDCs.

### KLF4 induces glycolytic shift in VSMCs.

Recent studies revealed that cell transformation (e.g., epithelial-mesenchymal transition)

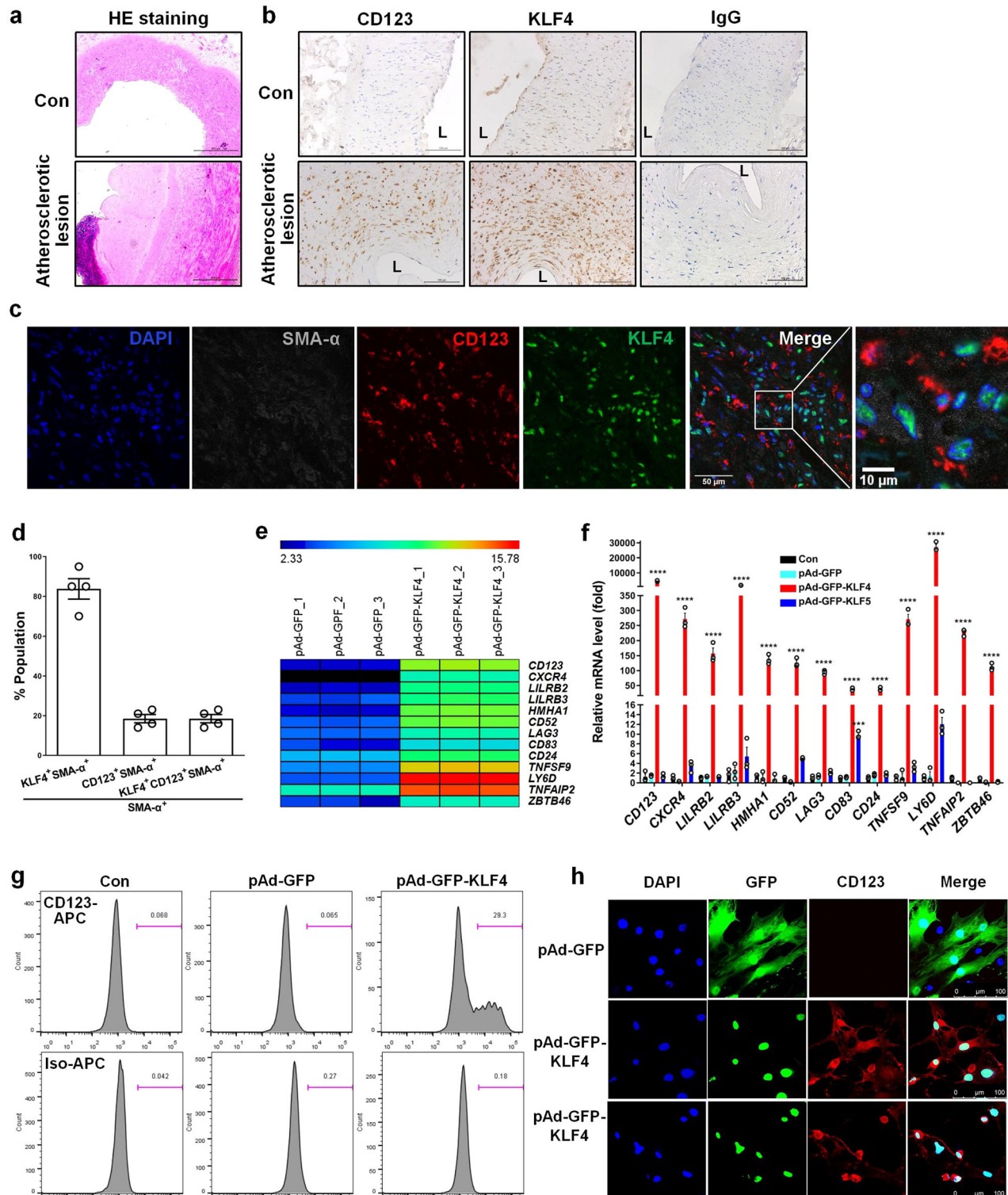

**Fig. 1 KLF4 plays a critical role in triggering pDC markers expression in VSMCs. a**, **b** Representative HE staining (bars = 500 μm) (**a**) and immunohistochemistry (IHC) for CD123 and KLF4 (bars = 100 μm) (**b**) of cross sections from human renal arteries with atherosclerotic lesions and normal renal arteries (Con). L = lumen. **c** Representative multiplex IHC staining of the core region of human renal artery atherosclerotic lesions. bars = 50 μm. **d** Quantification of the frequency of KLF4+SMA-α+, CD123+SMA-α+, and KLF4+CD123+SMA-α+ cells as a percent of total SMA-α+ cells in the core region of human renal artery atherosclerotic lesions (n = 4 independent experiments, error bars show SEM). **e** A subset of the differentially expressed mRNAs detected in VSMCs infected with pAd-GFP or pAd-GFP-KLF4 by using microarray analysis were selected and summarized (n = 3 independent samples per group). **f** mRNAs shown in **e** were determined by qRT-PCR. ***P < 0.005 and ****P < 0.001 vs pAd-GFP (n = 3 independent experiments, error bars show SEM). One-way ANOVA with Tukey's multiple comparison tests were performed. **g**, **h** Flow cytometry analysis (**g**) and immunofluorescent staining (**h**) of mock- and KLF4-transduced VSMCs. bars = 100 μm.

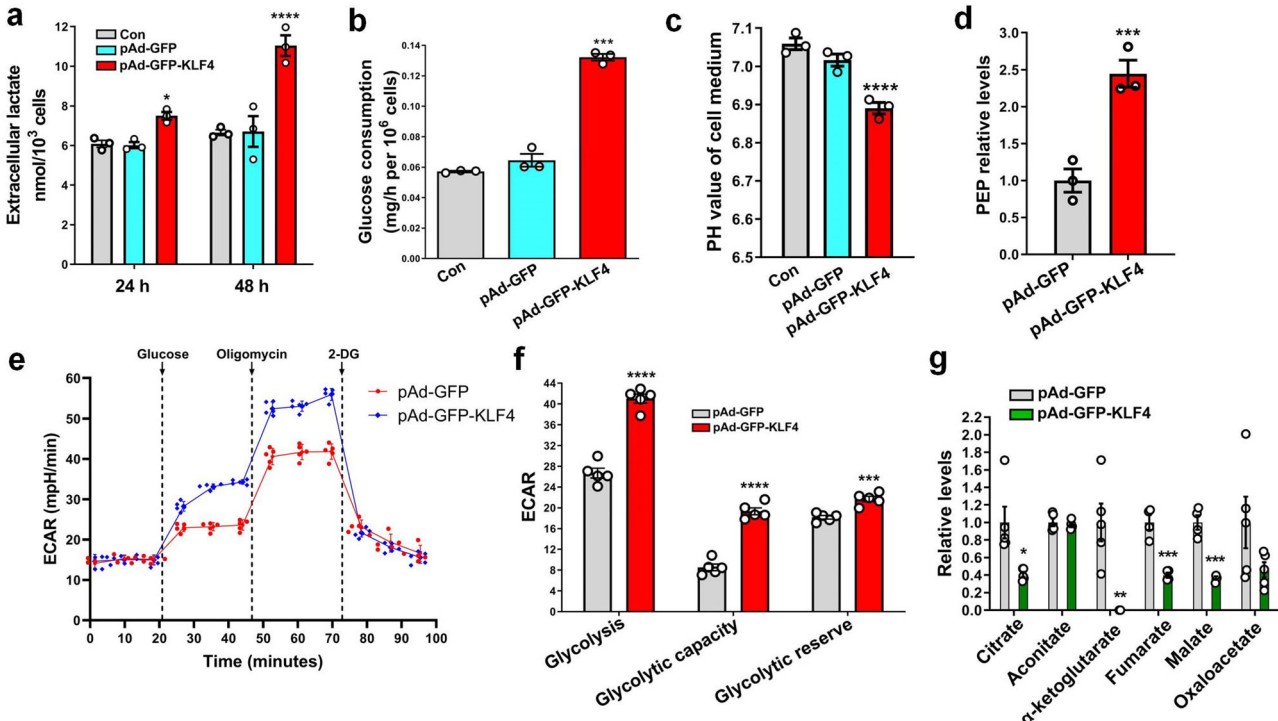

**Fig. 2 KLF4 induces glycolytic shift in VSMCs. a–d** Levels of lactate (**a**), glucose consumption (**b**), medium acidification (**c**) and phosphoenol pyruvic acid (PEP) (**d**) in VSMCs infected with pAd-GFP or pAd-GFP-KLF4. **e** ECAR profile showing glycolytic function in mock- and KLF4-transduced VSMCs measured by Seahorse glycolysis stress assay. Vertical lines indicate the time of addition of glucose (10 mM), oligomycin (1 μM), and 2-DG (50 mM). **f** Quantification of glycolytic function parameters of **e**. **g** Levels of TCA cycle intermediates in mock- and KLF4-transduced VSMCs. *$P < 0.05$, **$P < 0.01$, ***$P < 0.005$ and ****$P < 0.001$ vs pAd-GFP ($n = 3$ independent experiments for **a–d**, $n = 5$ independent samples for **f**, and $n = 4$ or 5 independent samples per group for **g**, error bars show SEM). One-way ANOVA with Tukey's multiple comparison tests were performed for **a–c**. Unpaired Student's $t$ tests were performed for **d**, **f**, **g**.

is accompanied by a glycolytic shift[18,19]. We were interested in determining if a glycolytic switch occurred during KLF4-induced phenotypic switching of VSMCs to pDC-like cells. Thus, we measured and compared glucose metabolic profiles between KLF4-overexpressing VSMCs and control cells. The results showed that the enforced expression of KLF4 in VSMCs for 24 or 48 h enhanced lactate production (Fig. 2a), glucose consumption (Fig. 2b), medium acidification (Fig. 2c) and the level of phosphoenol pyruvic acid (PEP), a glycolytic intermediate (Fig. 2d), compared with those of pAd-GFP-infected cells. To further confirm metabolic phenotype switches in VSMCs, we performed Seahorse glycolysis stress tests by Seahorse extracellular flux analysis. Similarly, we observed that KLF4 overexpression increased the extracellular acidification rate (ECAR) related to both glycolysis and glycolytic capacity (Fig. 2e, f). Moreover, the glycolytic reserve was also increased in KLF4-overexpressing cells (Fig. 2f). In contrast, tricarboxylic acid (TCA) cycle intermediates were largely reduced in KLF4-overexpressing VSMCs (Fig. 2g), consistent with a glycolytic shift. These data suggest that a glycolytic switch occurs during KLF4-induced phenotypic switching of VSMCs to pDC-like cells.

**KLF4 enhances glycolysis by upregulating PFKFB3 expression.** To clarify whether the glycolytic switch induced by KLF4 in VSMCs was associated with expression changes in glycolytic enzymes, we performed mRNA microarray analyses of KLF4-overexpressing VSMCs vs control cells. As a result, *GPI*, *TPI1*, *SDHA* and *PFKFB3* were upregulated, with a 2.0-, 2.3-, 2.6- and 6.4-fold increase, respectively, in KLF4-overexpressing VSMCs. Other metabolic genes showed no obvious changes (Fig. 3a).

Among these upregulated genes, only PFKFB3 protein level was upregulated by 7.2-fold in KLF4-overexpressing VSMCs relative to controls, as demonstrated by the TMT-based LC-MS/MS analysis (Fig. 3b). As well, Western blot analysis also confirmed that PFKFB3 but not HK2 (a key glycolytic enzyme) protein expression was largely induced by KLF4 overexpression (Fig. 3c–e). Unexpectedly, quantitative RT-PCR using two pairs of primers to amplify different regions of *PFKFB3* gene, respectively, showed that *PFKFB3* mRNA level was not affected by KLF4 overexpression (Fig. 3f). Because PFKFB3 functions mainly in producing fructose-2,6-bisphosphate (F2,6BP) that allosterically regulates the activity of the enzymes phosphofructokinase 1 (PFK-1) and fructose 1,6-bisphosphatase (FBPase-1) to promote glycolytic flux, we investigated whether KLF4 promoted glycolysis by inducing PFKFB3 protein expression. As expected, overexpression of PFKFB3 with a lentiviral vector encoding PFKFB3 (Fig. 3g) markedly increased lactate production (Fig. 3h). Further, PFKFB3 knockdown by short interfering RNA (si-PFKFB3) dramatically decreased lactate production and largely abrogated lactate elevation and CD123 expression elicited by KLF4 overexpression (Fig. 3i–k). We used another siRNA targeting PFKFB3 (si-PFKFB3#) to repeat the experiments and got the same results (Supplementary Fig. 3a–c). In addition, using TSA staining, we observed a multiple co-localization of PFKFB3, KLF4, and SMA-α in the human atherosclerotic lesion (Fig. 3l). Thus, PFKFB3 upregulation induced by KLF4 is at least in part responsible for the enhanced glycolysis in VSMCs.

**KLF4 upregulates PFKFB3 expression via eEF1A2.** The fact that KLF4 increases PFKFB3 protein levels but does not influence

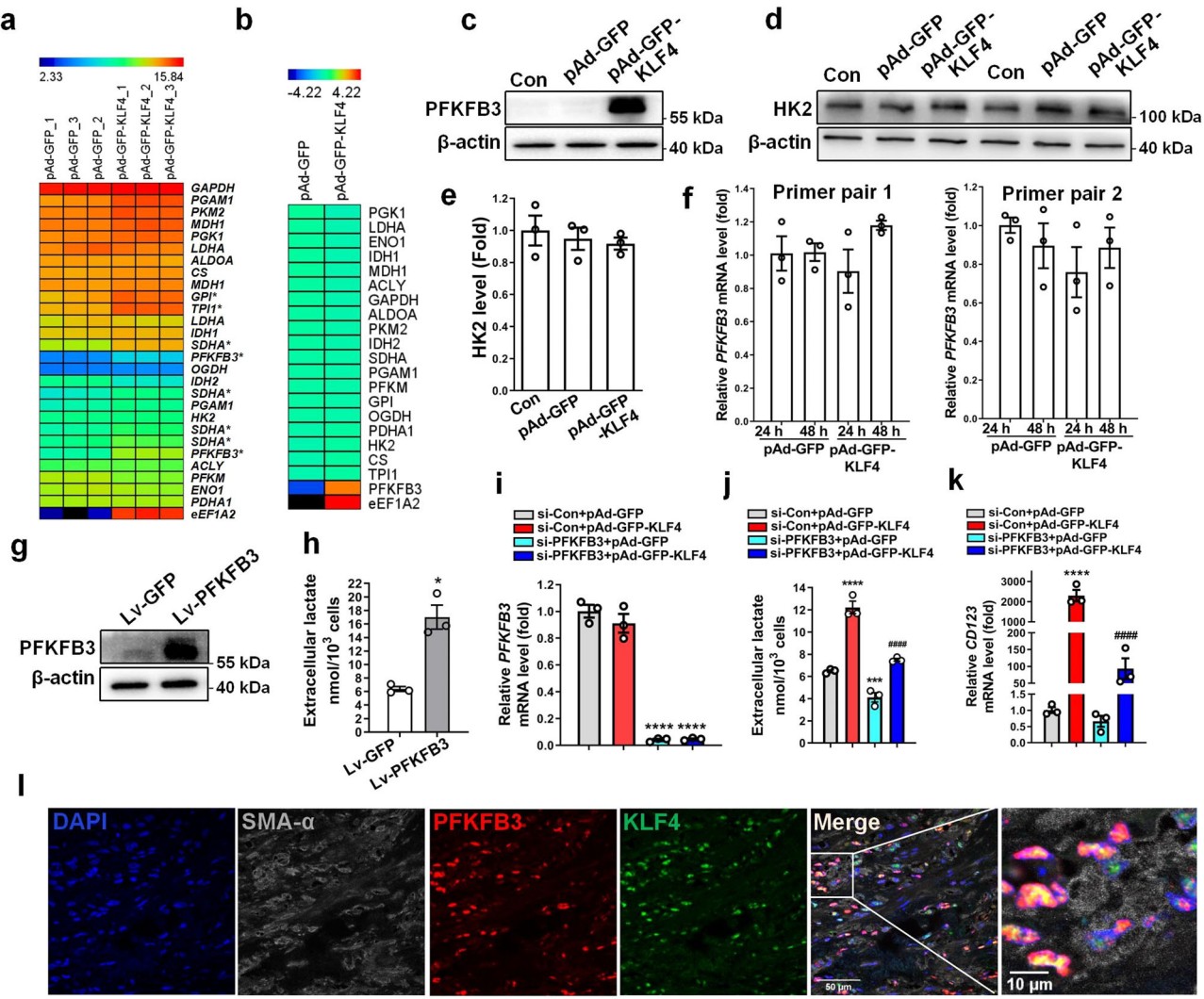

**Fig. 3 KLF4 enhances glycolysis by upregulating PFKFB3 expression. a, b** A subset of mRNAs (**a**) and proteins (**b**) detected in VSMCs infected with pAd-GFP or pAd-GFP-KLF4 (n = 3 independent samples per group for **a**, n = 1 independent sample for **b**) by using microarray analysis and TMT-based LC-MS/MS analysis, respectively, were selected and summarized. **c–f** VSMCs were infected with pAd-GFP or pAd-GFP-KLF4 as indicated. PFKFB3 and β-actin protein levels were detected by immunoblotting (**c**). HK2 protein levels were detected by immunoblotting (**d**) and quantified by normalizing to β-actin (**e**). PFKFB3 mRNA levels were assessed by qRT-PCR with two pairs of primers (**f**). **g, h** VSMCs were infected with Lv-GFP or Lv-PFKFB3. PFKFB3 and β-actin expression levels were measured by immunoblotting (**g**). Analysis of lactate production levels (**h**). *P < 0.05 vs Lv-GFP (n = 3 independent experiments, error bars show SEM). **i–k** VSMCs were transfected with the indicated constructs. PFKFB3 mRNA levels were assessed by qRT-PCR (**i**). Analysis of lactate production levels (**j**). CD123 mRNA levels were assessed by qRT-PCR (**k**). ***P < 0.005 and ****P < 0.001 vs si-Con+pAd-GFP, ####P < 0.001 vs si-Con+pAd-GFP-KLF4 (n = 3 independent experiments, error bars show SEM). One-way ANOVA with Tukey's multiple comparison tests were performed for **e**, **f**, **i–k**. Unpaired Student's t tests were performed for **h**. **l** Representative multiplex IHC staining of the core region of human renal artery atherosclerotic lesions. Bars = 50 μm.

its mRNA expression implies that KLF4 might regulate PFKFB3 stability post-transcriptionally. To test this, we used protein synthesis inhibitor cycloheximide (CHX) to block de novo protein biosynthesis to probe the effect of KLF4 over-expression on PFKFB3 stability in VSMCs. Unexpectedly, KLF4 overexpression did not affect the half-life of PFKFB3 protein (Fig. 4a). To determine the mechanisms underlying the increased protein level of PFKFB3 by KLF4 overexpression, we compared the gene expression profiles and the TMT-based LC-MS/MS data between KLF4-overexpressing VSMCs and control cells. We found that eEF1A2, an isoform of the alpha subunit of the eukaryotic elongation factor-1 complex (eEF1A), was largely upregulated by KLF4 overexpression at both mRNA and protein levels (Fig. 3a, b). This finding was further confirmed by quantitative RT-PCR and Western blot analysis (Fig. 4b, c). This prompted us to investigate whether KLF4 increases PFKFB3

protein levels at the translation level via inducing eEF1A2 expression. To obtain a global view of the upregulated proteins by eEF1A2, we performed a TMT-based LC-MS/MS analysis of GFP- and eEF1A2-overexpressing VSMCs. As a result, 40 over-lapping upregulated proteins were obtained by KLF4 and eEF1A2 overexpression in VSMCs. Among them, only 2 proteins were simultaneously upregulated more than 2-fold, 38 proteins had a 1.2-fold increase (Fig. 4d). Of note, although PFKFB3 protein level was not upregulated by eEF1A2 overexpression (Fig. 4d, e), knockdown of eEF1A2 by two short interfering RNAs (si-eEF1A2 and si-eEF1A2#) largely blocked KLF4-induced PFKFB3 upre-gulation (Fig. 4f and Supplementary Fig. 4), indicating that eEF1A2 is necessary for PFKFB3 protein expression induced by KLF4.

Based on the above findings, we hypothesized that a coactivator for eEF1A2, which can be induced by KLF4, might be required to

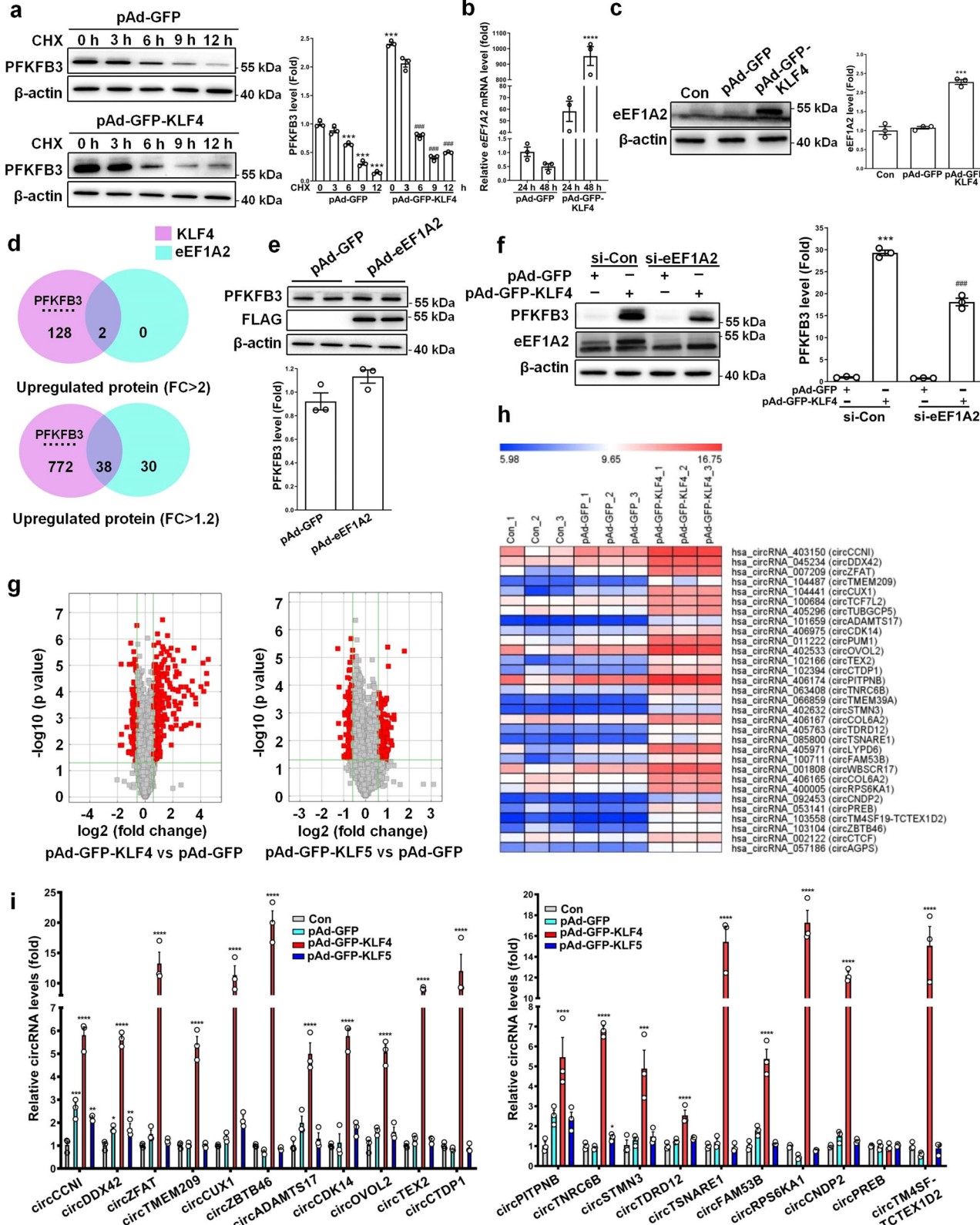

effectively activate PFKFB3 translation. Thus, we focused on circular RNA (circRNA), a class of noncoding RNAs involved in various cell functions. We first used circRNA microarrays to compare circRNA profiles between KLF4-overexpressing VSMCs and control cells. As a result, a total of 486 circRNAs were found to be differentially expressed with fold change ≥1.5 in expression level. Among them, 287 and 199 circRNAs were upregulated and

downregulated, respectively, in KLF4-overexpressing VSMCs (Fig. 4g). The enforced expression of KLF5 in VSMCs did not induce the changes of circRNA expression profiles similar to those induced by KLF4 (Fig. 4g). To further validate the microarray data, 31 upregulated circRNAs (Fig. 4h) by KLF4 (fold change>4) were selected to verify their expression in KLF4-overexpressing VSMCs. Quantitative RT-PCR showed that 20

**Fig. 4 KLF4 upregulates PFKFB3 expression via eEF1A2. a** VSMCs infected with pAd-GFP or pAd-GFP-KLF4 were treated with 20 μg/ml of cycloheximide (CHX) for the indicated times. PFKFB3 protein levels were detected by immunoblotting and quantified by normalizing to β-actin. ***$P < 0.005$ vs CHX 0 h+pAd-GFP, ###$P < 0.005$ vs CHX 0 h+pAd-GFP-KLF4 ($n = 3$ independent experiments, error bars show SEM). **b** *eEF1A2* mRNA levels were assessed by qRT-PCR. ****$P < 0.001$ vs pAd-GFP ($n = 3$ independent experiments, error bars show SEM). **c** eEF1A2 protein levels in mock- and KLF4-transduced VSMCs were measured by immunoblotting and quantified by normalizing to β-actin. ***$P < 0.005$ vs pAd-GFP ($n = 3$ independent experiments, error bars show SEM). **d** A Venn diagram showing 40 overlapping proteins between KLF4- and eEF1A2-overexpressing VSMCs identified by the TMT-based LC-MS/MS analysis. Two proteins were simultaneously upregulated more than 2-fold by KLF4 and eEF1A2 (upper), and 38 proteins had a 1.2-fold increase (lower). **e** VSMCs were infected with pAd-GFP or pAd-eEF1A2 with a flag tag. PFKFB3 protein levels were measured by immunoblotting and quantified by normalizing to β-actin ($n = 3$ independent experiments, error bars show SEM). **f** VSMCs were transfected with the indicated constructs. PFKFB3, eEF1A2 and β-actin protein levels were measured by immunoblotting and PFKFB3 protein levels were quantified by normalizing to β-actin. ***$P < 0.005$ vs si-Con+pAd-GFP, ###$P < 0.005$ vs si-Con+pAd-GFP-KLF4 ($n = 3$ independent experiments, error bars show SEM). **g, h** circRNA microarrays were performed in VSMCs treated as indicated. Volcano plots are used for visualizing differentially expressed circRNAs between the two groups (fold change>1.5, $P < 0.05$, $n = 3$ independent samples each group). The red blocks indicate differentially expressed circRNAs; gray blocks indicate circRNAs with no difference in their expression (**g**). 31 upregulated circRNAs by KLF4 (fold change>4) were selected and summarized (**h**). **i** qRT-PCR detected the indicated circRNAs in VSMCs treated as indicated. *$P < 0.05$, **$P < 0.01$, ***$P < 0.005$, and ****$P < 0.001$ vs pAd-GFP ($n = 3$ independent experiments, error bars show SEM). One-way ANOVA with Tukey's multiple comparison tests were performed for **a–c**, **f**, **i**. Unpaired Student's *t* tests were performed for **e**.

circRNAs were dramatically upregulated by KLF4, but not by KLF5 overexpression (Fig. 4i). We next successfully constructed 15 circRNA-expressing plasmids and confirmed that transfecting VSMCs with these recombined plasmids increased their corresponding circRNA levels (Supplementary Fig. 5).

**eEF1A2 interacts with circCTDP1 to regulate PFKFB3.** Although the functions of most circRNAs remain largely unknown, some circRNAs are well known to play important roles in different cell functions by distinct modes of action[26]. Here, we sought to know whether eEF1A2 could interact cooperatively with a circRNA, as a coactivator for eEF1A2, to regulate PFKFB3 protein expression. Thus, eEF1A2-overexpressing VSMCs were transfected with various circRNA expression plasmids indicated as in Supplementary Fig. 5. Strikingly, two circRNAs, circZFAT (#1) and circCTDP1 (#2), were found to increase PFKFB3 protein levels when co-overexpressed with eEF1A2 (Fig. 5a and Supplementary Fig. 6). The presence of circZFAT and circCTDP1 was confirmed using divergent primers to amplify circRNAs formed by head-to-tail splicing (Supplementary Fig. 7). To further confirm circZFAT or circCTDP1 cooperation with eEF1A2 to regulate PFKFB3, we designed 2 siRNAs: one siRNA targeting the back-splice sequence of two circRNAs (si-circZFAT or si-circCTDP1), and another targeting the sequence shared by both the linear and circular transcripts (si-both) (Fig. 5b). Transfecting VSMCs with si-circZFAT or si-circCTDP1 knocked down only their corresponding circRNA but did not affect their mRNA levels, whereas si-both knocked down not only their circRNAs but also their mRNA expression (Fig. 5c). Importantly, knockdown of circCTDP1 by si-circCTDP1 considerably abrogated KLF4-induced PFKFB3 upregulation, while knockdown of circZFAT did not (Fig. 5d). Similar results were obtained when circCTDP1 was knocked down by si-both (Fig. 5e). Additionally, simultaneous knockdown of eEF1A2 and circCTDP1 strongly blocked KLF4-induced PFKFB3 upregulation, with PFKFB3 expression returning to the control levels (Fig. 5f). We used different siRNAs (named si-circCTDP1#, si-circZFAT#, si-both-C#, and si-both-Z#, respectively) to repeat the experiments and got the same results (Supplementary Fig. 8). Collectively, these data clearly suggest that eEF1A2 enhances PFKFB3 protein expression in a circCTDP1-dependent manner.

We next examined whether eEF1A2 interacted with circCTDP1 in VSMCs by RNA-binding protein immunoprecipitation (RIP) with an antibody to eEF1A2. The results showed that in KLF4-overexpressing VSMCs, circCTDP1 was enriched by at least 10-fold in the anti-eEF1A2 immunoprecipitates compared to those in anti-IgG immunoprecipitates (Fig. 5g). Further, we used a specific DNA probe for circCTDP1 sequence (circCTDP1 probe) to pull down the protein interacting with circCTDP1 and then detected circCTDP1 and eEF1A2 in the precipitates by quantitative RT-PCR and Western blotting, respectively. As expected, the circCTDP1 probe could specifically bind to circCTDP1, with circCTDP1 being enriched by more than 10-fold in KLF4-overexpressing VSMCs (Fig. 5h), and this probe also effectively pulled down eEF1A2 relative to the control probe (NC probe) (Fig. 5i). In addition, using a combined RNA in situ hybridization/protein staining, we observed a physical co-localization of circCTDP1 and eEF1A2 in VSMCs (Fig. 5j). Using computer algorithm, we predicted probable RNA-binding residues in eEF1A2 and nucleotides interacting with eEF1A2 in circCTDP1 sequence (Supplementary Fig. 9a). The secondary and tertiary structures of circCTDP1 were analyzed as described previously[27]. NPDock was used to perform the in silico molecular docking between circCTDP1 and eEF1A2. The results showed that eEF1A2 could dock into one circCTDP1 (Supplementary Fig. 9b, c). These results suggest that eEF1A2 interacts specifically with circCTDP1 in VSMCs.

**Glycolytic switch is required for the phenotypic switching of VSMCs to pDC-like cells.** In the present study, KLF4 promotes glycolysis by upregulating the key glycolytic enzyme PFKFB3. Next, we sought to determine if the glycolytic shift is required for VSMC phenotypic switching to pDC-like cells. To do this, we used hexokinase (HK) inhibitor 2-deoxy-D-glucose (2-DG) and PFK15 [1-(4-pyridinyl)-3-(2-quinolinyl)-2-propen-1-one24], a selective inhibitor of PFKFB3, to inhibit glycolysis, respectively. When inhibiting glycolytic enzyme activities, the two inhibitors did not affect PFKFB3 expression (Supplementary Fig. 10a). Consistent with the loss-of-function experiment with siRNA vs PFKFB3 in Fig. 3k, quantitative RT-PCR showed that 2-DG and PFK15 dose-dependently suppressed KLF4-induced CD123 expression (Fig. 6a, b).

Because previous studies have indicated that many signaling molecules and pathways appear to have a role in VSMC phenotypic switching induced by lactate[28], and that STAT3 activation (Tyr705 phosphorylation) is essential for CD123 expression in mouse leukemia cells[29], we sought to determine which signaling pathways are involved in glycolysis-induced phenotypic switching of VSMCs. First, we examined the effects of KLF4-induced glycolytic switch on STAT3 and Akt activation. Immunoblotting revealed that STAT3 phosphorylation at Ser727, but not at Tyr705, as well as acetylation at Lys685 were enhanced

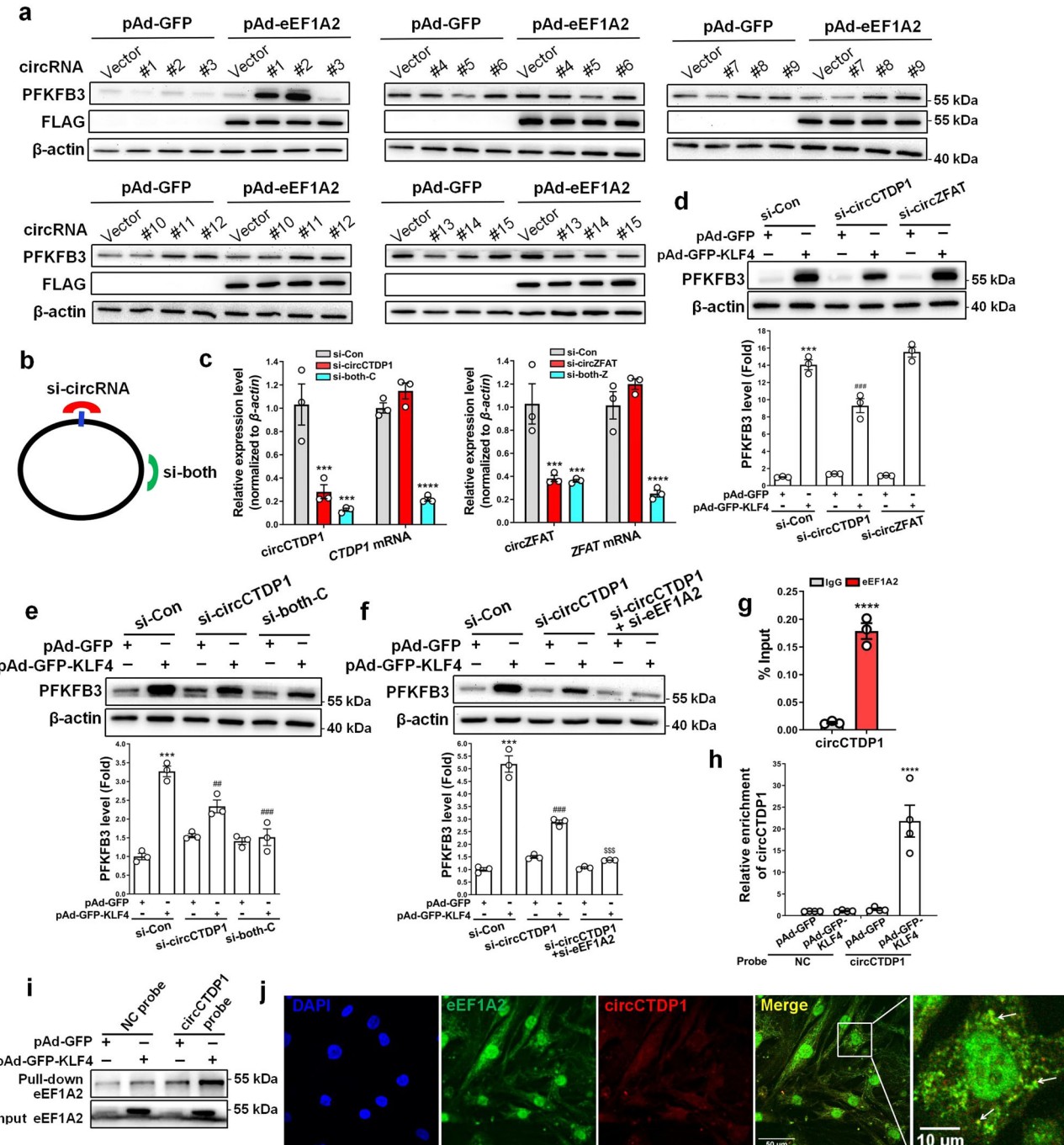

**Fig. 5 eEF1A2 interacts with circCTDP1 to regulate PFKFB3. a** VSMCs were transfected with the indicated constructs. PFKFB3, flag and β-actin protein levels were measured by immunoblotting. #1 to 15 represent constructs of circZFAT, circCTDP1, circCUX1, circRPS6KA1, circTM4SF-TCTEX1D2, circTSNARE1, circDDX42, circTNRC6B, circZBTB46, circADAMTS17, circTMEM209, circCNDP2, circTEX2, circFAM53B-1, and circFAM53B-2, respectively. **b** Schematic illustration showing two targeted siRNAs. si-circRNA targets the back-splice junction of circCTDP1 or circZFAT, and si-both targets both the linear and circular transcripts. **c** qRT-PCR detected the indicated circRNAs and mRNAs in VSMCs transfected with the two siRNAs shown in **b**. ***P < 0.005 and ****P < 0.001 vs si-Con (n = 3 independent experiments, error bars show SEM). **d**–**f** VSMCs were transfected with the indicated constructs. PFKFB3 protein levels were measured by immunoblotting and quantified by normalizing to β-actin. ***P < 0.005 vs si-Con+pAd-GFP, ##P < 0.01 and ###P < 0.005 vs si-Con+pAd-GFP-KLF4, $$$P < 0.005 vs si-circCTDP1+pAd-GFP-KLF4 (n = 3 independent experiments, error bars show SEM). **g** RNA immunoprecipitation (RIP) was performed with anti-IgG or anti-eEF1A2 antibody in lysates of VSMCs infected with pAd-GFP-KLF4, and then the immunoprecipitates were used to detect circCTDP1 by qRT-PCR. ****P < 0.001 vs IgG (n = 3 independent experiments, error bars show SEM). **h, i** RNA pull-down assay was performed in VSMCs infected with pAd-GFP or pAd-GFP-KLF4. Cell lysates were pulled down with probe against circCTDP1. qRT-PCR detected circCTDP1 (**h**). ****P < 0.001 vs pAd-GFP (n = 4 independent experiments, error bars show SEM). Western blotting detected eEF1A2 (**i**). **j** Representative immunofluorescent staining of eEF1A2 (green), circCTDP1 (red) and DAPI (blue) in VSMCs (bars = 50 μm). White arrow showed the co-localization between eEF1A2 and circCTDP1. Unpaired Student's t tests were performed for **c** and **g**. One-way ANOVA with Tukey's multiple comparison tests were performed for **d**–**f**, **h**.

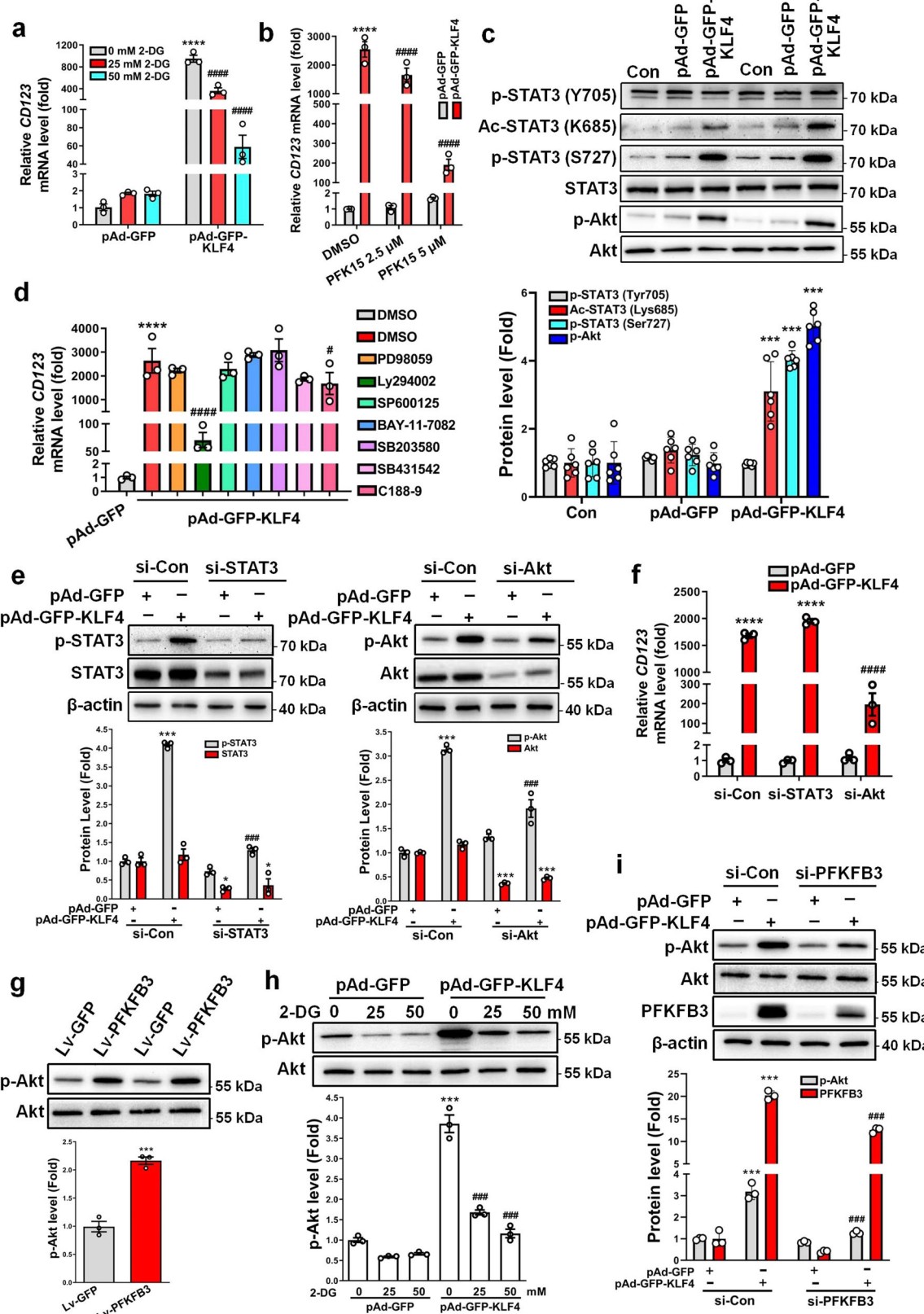

by KLF4 overexpression in VSMCs (Fig. 6c). However, inhibition of STAT3 activation by its inhibitor, C188-9, only resulted in a modest decrease in *CD123* mRNA expression compared with the vehicle control (Fig. 6d). This might imply that the effect of STAT3 activation on CD123 expression is cell-context dependent. More importantly, Akt phosphorylation was also largely

increased in KLF4-overexpressing VSMCs (Fig. 6c), and inhibition of Akt activation by treatment with Ly294002 dramatically blocked the upregulation of CD123 expression induced by KLF4 (Fig. 6d). Consistent with these observations, knockdown of Akt markedly decreased KLF4-induced CD123 expression, while knockdown of STAT3 did not (Fig. 6e, f). Further, we confirmed

**Fig. 6 Glycolytic switch is required for the phenotypic switching of VSMCs to pDC-like cells. a** VSMCs infected with pAd-GFP or pAd-GFP-KLF4 were treated with the indicated doses of 2-DG (2-deoxy-D-glucose). *CD123* mRNA levels were assessed by qRT-PCR. ****$P < 0.001$ vs pAd-GFP + 0 mM 2-DG, ####$P < 0.001$ vs pAd-GFP-KLF4 + 0 mM 2-DG ($n = 3$ independent experiments, error bars show SEM). **b** VSMCs infected with pAd-GFP or pAd-GFP-KLF4 were treated with the indicated doses of PFK15 [1-(4-pyridinyl)-3-(2-quinolinyl)-2-propen-1-one24]. *CD123* mRNA levels were assessed by qRT-PCR. ****$P < 0.001$ vs pAd-GFP + DMSO, ####$P < 0.001$ vs pAd-GFP-KLF4 + DMSO ($n = 3$ independent experiments, error bars show SEM). **c** Expression levels for indicated proteins were measured by immunoblotting in mock- and KLF4-transduced VSMCs. p-STAT3 (Tyr705), p-STAT3 (Ser727), and Ac-STAT3 (Lys685) protein levels were quantified by normalizing to STAT3. p-Akt protein levels were quantified by normalizing to Akt. ***$P < 0.005$ vs pAd-GFP ($n = 6$ independent experiments, error bars show SEM). **d** VSMCs infected with pAd-GFP-KLF4 were treated with the indicated inhibitors. *CD123* mRNA levels were assessed by qRT-PCR. ****$P < 0.001$ vs pAd-GFP + DMSO, #$P < 0.05$ and ####$P < 0.001$ vs pAd-GFP-KLF4 + DMSO ($n = 3$ independent experiments, error bars show SEM). PD98059 is the ERK inhibitor, Ly294002 is the PI3K/Akt inhibitor, SP600125 is the JNK inhibitor, BAY-11-7082 is the NF-κB inhibitor, SB203580 is the p38 MAPK inhibitor, SB431542 is the ALK5/TGF-β type I receptor inhibitor, and C188-9 is the STAT3 inhibitor. **e, f** VSMCs were transfected with the indicated constructs. Expression levels for indicated proteins were measured by immunoblotting and quantified by normalizing to β-actin (**e**). *CD123* mRNA levels were assessed by qRT-PCR (**f**). *$P < 0.05$, ***$P < 0.005$ and ****$P < 0.001$ vs si-Con+pAd-GFP, ###$P < 0.005$ and ####$P < 0.001$ vs si-Con+pAd-GFP-KLF4 ($n = 3$ independent experiments, error bars show SEM). **g** p-Akt and Akt levels were measured by immunoblotting in mock- and PFKFB3-transduced VSMCs and p-Akt protein levels were quantified by normalizing to Akt. ***$P < 0.005$ vs Lv-GFP ($n = 3$ independent experiments, error bars show SEM). **h** VSMCs infected with pAd-GFP or pAd-GFP-KLF4 were treated with the indicated doses of 2-DG. p-Akt and Akt levels were measured by immunoblotting and p-Akt protein levels were quantified by normalizing to Akt. ***$P < 0.005$ vs pAd-GFP + 0 mM 2-DG, ###$P < 0.005$ vs pAd-GFP-KLF4 + 0 mM 2-DG ($n = 3$ independent experiments, error bars show SEM). **i** VSMCs were transfected with the indicated constructs. Expression levels for indicated proteins were measured by immunoblotting. p-Akt protein levels were quantified by normalizing to Akt and PFKFB3 protein levels were quantified by normalizing to β-actin. ***$P < 0.005$ vs si-Con+pAd-GFP, ###$P < 0.005$ vs si-Con+pAd-GFP-KLF4 ($n = 3$ independent experiments, error bars show SEM). One-way ANOVA with Tukey's multiple comparison tests were performed for **a–f**, **h**, **i**. Unpaired Student's *t* tests were performed for **g**.

that glycolytic switch induced by PFKFB3-enforced expression in VSMCs evidently increased Akt phosphorylation (Fig. 6g), whereas inhibition of glycolysis by 2-DG or PFKFB3 knockdown largely attenuated KLF4-induced Akt phosphorylation (Fig. 6h, i and Supplementary Fig. 10b) compared with their corresponding controls. Together, these findings suggest that KLF4 and PFKFB3 lie upstream of the PI3K/Akt signaling, and that glycolytic shift induced by KLF4 and PFKFB3 facilitates the phenotypic switching of VSMCs to pDC-like cells via activating Akt signaling.

**TMAO induces the phenotypic switching of VSMCs to a dysfunctional pDC-like cell by upregulating KLF4 expression.** Because TMAO, a gut-microbiota-dependent metabolite, has been shown to enhance atherosclerosis in animal models and is associated with cardiovascular risks in clinical studies[25], we sought to investigate the relationship between TMAO-induced atherosclerosis and KLF4-induced glycolytic switch in VSMCs. Using Western blotting, we found that TMAO increased KLF4 and PFKFB3 protein expression in a dose- and time-dependent manner (Fig. 7a, b). Simultaneously, TMAO also greatly increased the mRNA expression levels of *CD123*, *KLF4* and *eEF1A2* in a dose- and time-dependent manner, without affecting *PFKFB3* mRNA levels (Fig. 7c–j), consistent with the above-mentioned observations that KLF4 overexpression did not affect *PFKFB3* mRNA expression (Fig. 3f). More interestingly, circCTDP1 expression in VSMCs was also dramatically upregulated by treatment with TMAO (Fig. 7k). In another experiment, we found that treating VSMCs with TMAO decreased the expression of VSMC differentiation markers SM22α and SMα-actin (Supplementary Fig. 11a, b). Consistent with the data in Supplementary Fig. 2e, TMAO treatment could not induce IFN-α production (Supplementary Fig. 11c). These results indicated that TMAO could convert VSMC to a dysfunctional pDC-like cell.

To further determine the role of KLF4 in TMAO-induced VSMC phenotypic switching, we designed two siRNAs targeting KLF4, i.e., si-KLF4 and si-KLF4#, both of which were effective in lowering KLF4 protein level and thus were used in the subsequent experiments (Supplementary Fig. 11d). Silencing KLF4 largely eliminated the induced effects of TMAO on *CD123* and *eEF1A2* mRNA expression, whereas knockdown of PFKFB3 only attenuated TMAO-induced *CD123* mRNA expression but did

not impact *eEF1A2* mRNA level (Fig. 7l, m and Supplementary Fig. 11e, f). Further, knockdown of eEF1A2 or circCTDP1 alone resulted in a decrease in CD123 expression induced by TMAO, simultaneous knockdown of eEF1A2 and circCTDP1 further attenuated TMAO-induced CD123 expression (Fig. 7n and Supplementary Fig. 11g). To provide further evidence that KLF4 is required for TMAO-induced VSMC phenotypic switching, we knocked down KLF4 by si-KLF4 in VSMCs and observed the effects of TMAO on PFKFB3 and eEF1A2 protein expression. The results showed that silencing KLF4 largely suppressed TMAO-induced expression of PFKFB3 and eEF1A2 proteins (Fig. 7o and Supplementary Fig. 11h). Likewise, knockdown of eEF1A2 or circCTDP1 also obviously blocked PFKFB3 protein upregulation induced by TMAO (Fig. 7p and Supplementary Fig. 11i). Notably, although PDGF-BB, which is well known to induce VSMC phenotypic modulation, could upregulate *KLF4* and *eEF1A2* expression, it did not affect *CD123* mRNA and circCTDP1 expression (Fig. 7q–t). Taken together, these results strongly suggest that a KLF4-eEF1A2/circCTDP1-PFKFB3 regulatory axis is formed in the presence of TMAO to specifically induce VSMC glycolytic shift and phenotypic switching to pDC-like cells.

**TMAO enhances atherosclerosis by inducing VSMC phenotypic switching in Apoe$^{-/-}$ mice.** TMAO is known to enhance atherosclerosis in animal models[30]. We next investigated whether VSMCs underwent a phenotypic switching to a pDC-like phenotype in atherosclerotic lesions induced by TMAO. Wild-type (WT) and apolipoprotein E-null (Apoe$^{-/-}$) mice were fed with either a normal diet (contains 0.07–0.08% total choline, wt/wt) or a normal diet supplemented with high amounts of additional choline (1.3%) in the presence vs absence of 3,3-dimethyl-1-butanol (DMB, 1.0%, v/v, provided in the drinking water) which is shown to non-lethally reduce TMAO levels in mice fed a high choline diet. At 18 weeks of age, plasma TMAO levels were increased in mice fed diets supplemented with choline, and the addition of DMB resulted in a reduction in plasma TMAO levels in the choline-supplemented mice (Supplementary Fig. 12a). As expected, chronic exposure of Apoe$^{-/-}$ mice to a choline-supplemented diet resulted in an enhancement in atherosclerotic plaque formation near the arterial bifurcations and bends, and the

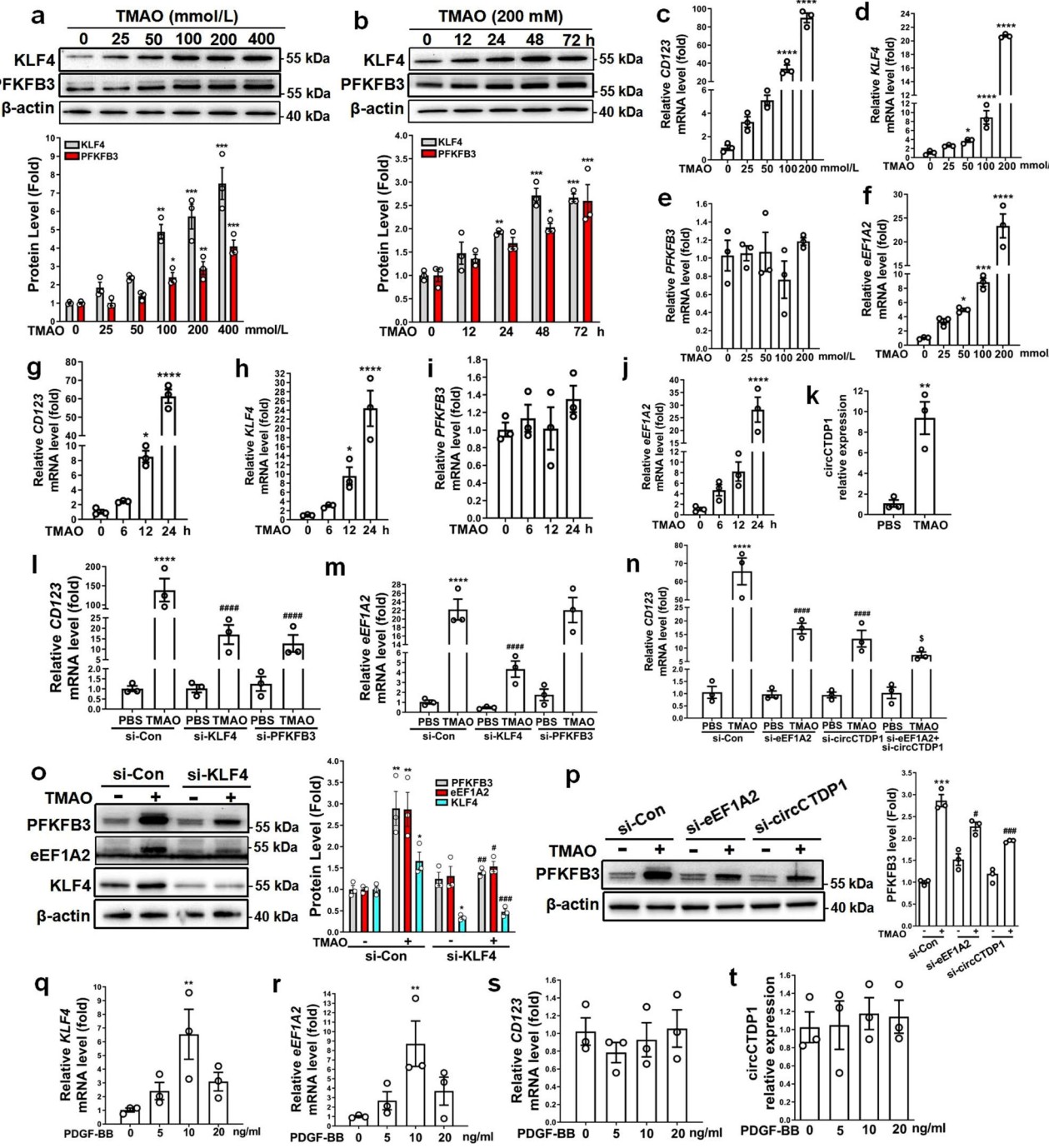

**Fig. 7 TMAO induces the phenotypic switching of VSMCs to a dysfunctional pDC-like cell by upregulating KLF4 expression. a–j** VSMCs were cultured in serum-free medium for 24 h, followed by treatment with the indicated doses of TMAO for the indicated times. KLF4 and PFKFB3 protein expression levels were measured by immunoblotting and quantified by normalizing to β-actin (**a**, **b**). Expression levels for indicated mRNAs were assessed by qRT-PCR (**c–j**). *$P < 0.05$, **$P < 0.01$, ***$P < 0.005$ and ****$P < 0.001$ vs TMAO-untreated group ($n = 3$ independent experiments, error bars show SEM). **k** VSMCs were incubated in serum-free medium for 24 h, followed by treatment with 200 mM TMAO for 24 h. circCTDP1 levels were assessed by qRT-PCR. **$P < 0.01$ vs PBS ($n = 3$ independent experiments, error bars show SEM). **l–p** VSMCs were treated as indicated. Expression levels for indicated mRNAs were assessed by qRT-PCR (**l–n**). Expression levels for the indicated proteins were measured by immunoblotting and quantified by normalizing to β-actin (**o**, **p**). *$P < 0.05$, **$P < 0.01$, ***$P < 0.005$ and ****$P < 0.001$ vs si-Con+PBS group, #$P < 0.05$, ##$P < 0.01$, ###$P < 0.005$ and ####$P < 0.001$ vs si-Con+TMAO group, $$P < 0.05$ vs si-circCTDP1+TMAO ($n = 3$ independent experiments, error bars show SEM). **q–t** VSMCs were incubated in serum-free medium for 24 h, followed by treatment with the indicated doses of PDGF-BB. Expression levels for the indicated mRNAs and circCTDP1 were assessed by qRT-PCR ($n = 3$ independent experiments, error bars show SEM). **$P < 0.01$ vs PDGF-BB-untreated group ($n = 3$ independent experiments, error bars show SEM). One-way ANOVA with Tukey's multiple comparison tests were performed for **a–j**, **l–t**. Unpaired Student's *t* tests were performed for **k**.

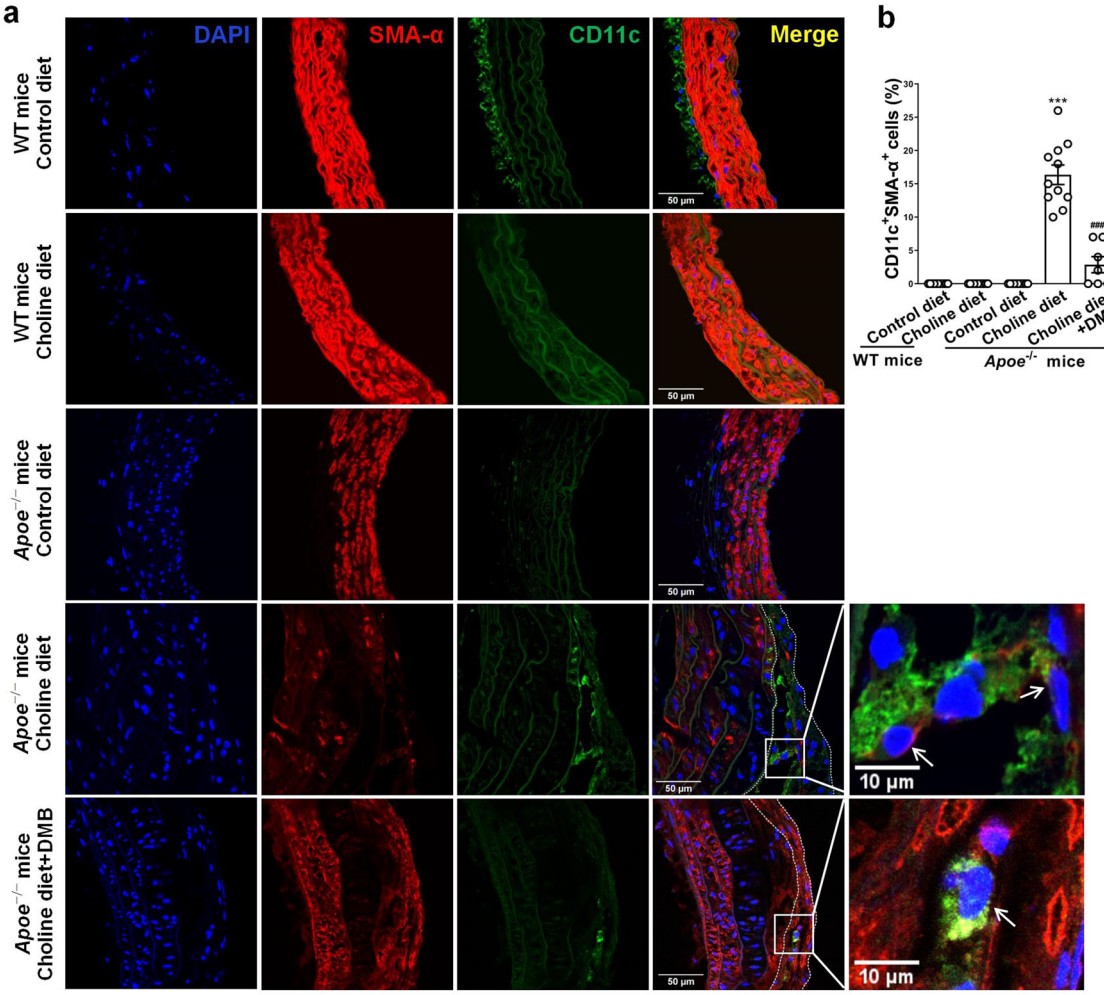

**Fig. 8 TMAO enhances atherosclerosis by inducing VSMC phenotypic switching in Apoe⁻/⁻ mice. a** WT or Apoe⁻/⁻ mice were fed with either a normal diet (contains 0.07–0.08% total choline, wt/wt) or a normal diet supplemented with high amounts of additional choline (1.3%) in the presence vs absence of DMB (1.0%, v/v, provided in the drinking water) for 14 weeks. Representative immunofluorescent staining of SMA-α (red), CD11c (green), and DAPI (blue) on cross-sections of the mouse aortic root (bars = 50 µm). White arrow showed the SMA-α⁺CD11c⁺ cells. **b** Quantification of the frequency of CD11c⁺SMA-α⁺ cells as a percent of total DAPI⁺ cells in the core region of atherosclerotic lesions. ***$P < 0.005$ vs control diet Apoe⁻/⁻ mice, ###$P < 0.005$ vs choline diet Apoe⁻/⁻ mice ($n = 7$ or 11 independent experiments, error bars show SEM). One-way ANOVA with Tukey's multiple comparison tests were performed.

addition of DMB markedly inhibited the choline diet-enhanced atherosclerosis (Supplementary Fig. 12b, c).

Further, confocal immunofluorescence staining showed that 16% of cells were dual positive for *mouse* DC marker CD11c and SM α-actin in the atherosclerotic lesions from aortic root of mice fed diets supplemented with choline (Fig. 8). In addition, we also observed that 16% of cells in the media were KLF4⁺SMA-α⁺ and 25% and 21% of cells in the atherosclerotic lesions were eEF1A2⁺SMA-α⁺ and PFKFB3⁺SMA-α⁺, respectively (Supplementary Fig. 12d–h). These double-positive cells were largely reduced in mice exposed to DMB (Fig. 8 and Supplementary Fig. 12d–h). Overall, these results suggest that increased TMAO by dietary choline supplementation enhances atherosclerosis and that a subset of VSMCs within atherosclerotic plaques express *mouse* DC marker CD11c.

## Discussion
VSMCs undergo phenotypic switching in response to pro-atherosclerotic stimuli or proinflammatory cytokines, forming VSMC-derived macrophage-like cells and SMC-derived MSC-like cells[5]. This switching can directly promote atherosclerosis by

having reduced ability to clear lipids, dying cells, and necrotic debris, as well as by exacerbating inflammation[2]. Since VSMC function can vary obviously depending on the nature of the phenotypic transitions, it is therefore very important to understand molecular mechanisms that may modulate this process. Here, we describe a role for KLF4-PFKFB3-driven glycolysis in VSMC phenotypic switching. To our knowledge, this is the first study to show a functional link between KLF4 and glycolytic shift in VSMCs.

The previous study has showed that VSMCs within atherosclerotic lesions express markers of macrophages, MSCs, and myofibroblasts[5]. pDC marker CD123-positive cells were also found to be co-localized considerably with VSMCs[7]. Consistently, we here also observed CD123 expression in human atherosclerotic plaques and a co-localization of CD123 with VSMCs (Fig. 1a–d and Supplementary Fig. 2). Importantly, over-expression of KLF4 dramatically induces CD123 expression in cultured VSMCs (Fig. 1e–h), further suggesting that VSMCs can switch into a pDC-like phenotype in KLF4-dependent manner. In response to cholesterol loading, Mϕ marker mac-2 is expressed at high levels in VSMCs through KLF4-dependent mechanisms[5,31].

However, overexpression of KLF4 does not increase any of the Mϕ-related genes such as CD68, mac-2, and ABCA1 expression as demonstrated by our mRNA microarray analysis. This implies that a coactivator for KLF4, which can be activated by cholesterol, might be required to effectively induce mac-2 expression, or that cholesterol loading-mediated KLF4 modification alteration is required for this gene activation. A previous study showed that gene expression and functional properties of VSMC-derived Mϕ-like cells are distinctly different from classical Mϕs, and these cells have reduced phagocytic capacity compared with activated peritoneal Mϕs[32]. Similarly, VSMC-derived pDC-like cells hardly have IFN-α-producing activities of the authentic pDCs, even stimulated with CpG oligodeoxynucleotides (CpG ODN) (Supplementary Fig. 2e). What the function of the phenotypically modulated cells is and how their functional properties affect atherogenesis are of great challenge and need further studies in the future.

Besides malignant cells, aerobic glycolysis is also a characteristic of nonmalignant proliferating cells and have been observed in adventitial fibroblasts[33], endothelial cells (ECs)[34], Mϕs[35] and VSMCs[36]. Importantly, recent studies revealed that there is an intrinsic link between a glycolytic switch and cell transdifferentiation[19] and reprogramming[16,17]. Here, we demonstrate that a glycolytic shift is required for phenotypic switching of VSMCs to pDC-like cells.

First, we found that glycolysis is enhanced and TCA cycle metabolites are reduced in KLF4-overexpressing VSMCs, indicating a glycolytic shift driven by KLF4 (Fig. 2). Second, KLF4 enhances glycolysis by upregulating PFKFB3 protein expression (Fig. 3). PFKFB3 is the first metabolic enzyme to be identified with an AUUUA instability element in its 3'-untranslated region[23], conferring instability and enhanced translational activity on its mRNA[37]. Our findings suggest that KLF4 can considerably increase PFKFB3 protein levels, but does not affect its mRNA levels and protein stability, suggesting that KLF4 may increase PFKFB3 expression at the translation level. Regarding the mechanism for KLF4-induced PFKFB3 expression, we show here that KLF4 can dramatically induce the expression of eEF1A2, an isoform of the alpha subunit of the eukaryotic elongation factor-1 complex (Figs. 3a, b and 4b, c). The latter plays a key role in delivering the aminoacylated-tRNA to the A site of the ribosome for decoding of mRNA by codon–anticodon interactions[38]. Using loss- and gain-of-function experiments, we found that knockdown of eEF1A2 blocks KLF4-induced PFKFB3 upregulation, but overexpression of eEF1A2 is unable to upregulate PFKFB3 protein levels (Fig. 4d–f). Considering that only 2 proteins, including eEF1A2, are upregulated more than 2-fold by eEF1A2 in eEF1A2-overexpressing VSMCs (Fig. 4d), we hypothesized that a coactivator for eEF1A2, which can be induced by KLF4, might be required to effectively enhance PFKFB3 translation. Because circRNA was recently found to exert functions as a platform for RNA-binding proteins[27,39,40], we focused the coactivator for eEF1A2 on circular RNA (circRNA). We used circRNA microarrays to screen differentially expressed circRNAs regulated by KLF4 (Fig. 4g–i) and identified a circRNA, circCTDP1, which can interact with eEF1A2 to cooperatively upregulate PFKFB3 protein levels (Fig. 5). To our knowledge, this is the first evidence that eEF1A2 can highly enhance protein expression at the translation level in a circRNA-dependent manner. Despite the recent advances in our understanding of circRNA functions, the excise mechanisms underlying the regulation of PFKFB3 expression by circCTDP1–eEF1A2 interaction remain to be further elaborated.

Glycolytic shift driven by KLF4 and PFKFB3 facilitates the phenotypic switching of VSMCs to pDC-like cells via activating Akt signaling (Fig. 6). Recent studies show that Akt activation stimulates the metabolic shift to glycolysis[41]. We show here that glycolytic shift induced by KLF4 and PFKFB3 can also activate Akt. Thus, Akt activation and glycolysis may form a positive feedback loop to regulate the phenotypic switching of VSMCs. Although a recent study demonstrates that lactate promotes synthetic phenotype in VSMCs, lactate does not activate Akt[28]. Thus, the mechanism underlying Akt activation by glycolysis may not be associated with increased lactate production and needs to be further elucidated.

Growing evidence suggests that TMAO is not only a novel biomarker for human cardiovascular disease, but is also a promoter of atherothrombotic diseases[25,42,43]. Mechanistically, TMAO partially contributes to the development of atherosclerosis by upregulating macrophage scavenger receptors and thus increasing foam cell formation[30]. Recently, TMAO is shown to act directly on ECs and VSMCs to promote vascular inflammation[44,45]. However, whether and how TMAO is involved in phenotypic switching of VSMCs remain largely unknown. In this study, we show that TMAO can induce KLF4 expression, thus leading to the phenotypic switching of VSMCs to pDC-like cells (Fig. 7). TMAO is well known to be a protein stabilizer[46]. Although TMAO upregulates KLF4 protein expression, it seems that TMAO is unable to stabilize KLF4 protein directly, as evidenced by the fact that KLF4 mRNA is also upregulated by TMAO (Fig. 7). Thus, it is likely that TMAO may upregulate KLF4 expression by stabilizing some other transcription factors that activate KLF4 transcription. We further revealed that VSMCs undergo phenotypic switching to a pDC-like phenotype in atherosclerotic lesions induced by TMAO in Apoe$^{-/-}$ mice fed a high choline diet (Fig. 8). Despite the lack of understanding of the molecular sensor for TMAO, these and other studies suggest that TMAO can signal to cells directly[44]. According to our results of the concentration-course experiments, 200 mmol/L TMAO was used in time-course and mechanism studies in vitro. This concentration of TMAO (mM) is more consistent with that used in Zheng's research on ECs[47], but is much higher than that in human plasma (μM)[48]. Atherosclerosis is a complicated chronic disease that develops over many years, and one can assume that lower concentrations of TMAO would have some impact on atherogenesis. Further, TMAO is not the only contributor to atherosclerosis in vivo, it may act together with other factors to promote VSMC phenotypic switching and atherosclerosis. Besides, there may be higher concentrations of TMAO accumulating in the location of atherosclerotic lesions, although it may be hard to detect experimentally.

A KLF4-eEF1A2/circCTDP1-PFKFB3 regulatory axis is formed in the presence of TMAO to specifically induce the glycolytic shift and subsequent phenotypic switching of VSMCs. Our study uncovers a new molecular mechanism underlying the phenotypic switching of VSMCs, showing a functional link between KLF4 and glycolytic shift during VSMC phenotypic switching. Future work will be necessary to fully elucidate the functional properties of the phenotypically modulated VSMCs and the mechanism of glucose metabolism regulation by intestinal microbiota metabolism, which will be helpful to develop novel therapeutic strategies for the treatment of atherosclerosis.

## Methods

**Human tissue harvest**. Human vascular samples were obtained from thirteen patients, seven with hypertension and six without hypertension. Patients who had hypertension at least 10 years managed blood pressure by using hypotensor. The renal arteries used in this study were obtained from 2015 to 2017 at the second hospital of Hebei Medical University (Shijiazhuang, China). The protocols for human studies were approved by the ethics committee of the Second Hospital of Hebei Medical University. Each of the surgical patients gave informed consent before donating tissue. Human renal arteries were fixed overnight in 10% neutral buffered formalin and processed for routine embedding in paraffin.

**Animal experiments**. All animal studies were approved by the Institutional Animal Care and Use Committee of Hebei Medical University (approval ID: HebMU 20080026) and all efforts were made to minimize suffering. 4-week-old male wild-type (WT) C57BL/6J mice or apolipoprotein E-null (Apoe$^{-/-}$) C57BL/6J mice were fed with either a normal diet (contains 0.07-0.08% total choline, wt/wt) or a normal diet supplemented with high amounts of additional choline (1.3%) in the presence vs absence of 3,3-dimethyl-1-butanol (DMB, 1.0%, v/v, provided in the drinking water). Choline content of all diets was confirmed by LC/MS/MS. After 14 weeks, all mice were anesthetized and fasting blood was collected for measuring TMAO. Then mice were perfused with cold saline, and the arteries were harvested for lipid analyses and immunostaining.

**Cell culture and treatment**. Human aortic smooth muscle cells (HASMCs) (ScienCell, no. 6110) were grown in Smooth Muscle Cell Medium containing apo-transferrin, insulin, fibroblast growth factor-2, insulin-like growth factor-1, hydrocortisone, and 2% fetal bovine serum (FBS) (ScienCell, no. 1101). Cells were maintained in 5% $CO_2$ at 37 °C within a humidified atmosphere and determined to be SMCs by morphology and expression of SMα-actin. For the gain-of-function experiments, cells were transfected with adenoviruses or plasmids for 48 h unless otherwise indicated. For the loss-of-function experiments, cells were transfected with siRNAs for 48 h unless otherwise indicated. Before stimulation with TMAO (Sigma, no. 317594) or PDGF-BB (Proteintech, no. HZ-1308) and infection with adenoviruses (pAds), HASMCs were incubated in serum-free medium for 24 h. The duration of treatment with TMAO or PDGF-BB was 24 h unless otherwise indicated.

**Virus expression vector and plasmid constructs**. Adenoviruses encoding GFP (pAd-GFP), KLF4 (pAd-GFP-KLF4), KLF5 (pAd-GFP-KLF5), and eEF1A2 (pAd-eEF1A2-FLAG) and Lentivirus encoding PFKFB3 (Lv-PFKFB3) and GFP (Lv-GFP) were entrusted to Hanbio, shanghai. In order to overexpress a circular RNA with a seamless connection between its 5' and 3' ends without redundant nucleotides, we constructed a circ-pcDNA3.1 vector with a reverse repeat sequence combined with 5' donor splice sequences and 3' acceptor splice sequences (Supplementary Table 1). Between the donor splice site and acceptor splice site, we inserted an EcoNI enzyme site with sequences of 5'-CCTCAG$^\vee$CTAGG-3' and a PmlI enzyme site with sequences of 5'-CAC$^\vee$GTG-3'. The full length sequence of circRNA was amplified from cDNA and inserted to EcoNI (NEB) and PmlI (NEB)-digested circ-pcDNA3.1 with one-step cloning (C112-02; Vazyme Biotech Co., Ltd.). All plasmids were sequenced for confirmation. All primers for plasmid construction are listed in Supplementary Table 2.

**Lipid analyses**. Mouse carotid artery was isolated and placed in 10% formalin for fixation overnight. Then the carotid artery was dehydrated in 30% sucrose at 4 °C overnight and embedded in OCT compound (Sakura Finetek USA, lnc.). Serial frozen sections (4 μm) of the carotid artery were obtained. Fifteen to twenty slides with four sequential sections each were prepared from each carotid artery, and two slides per carotid artery were processed for Oil Red O staining. For enface analysis, the entire aorta from the aortic root through the bifurcation of the iliac arteries was stained with Oil Red O for 2 h at room temperature, adjoining tissues were removed, and the aorta was opened longitudinally and pinned onto a white silicon bed. Images were captured by a camera (EOS 600D, Canon).

**Morphology analysis**. Human renal arteries were fixed overnight in 10% neutral buffered formalin and processed for routine embedding in paraffin. Ten consecutive 4-μm-thick sections were prepared for hematoxylin and eosin (HE) staining. Images were acquired using a Leica microscope (Leica DM6000B, Switzerland).

**Immunostaining**. Paraffin cross-sections (4-μm thick) from human renal arteries were deparaffinized with xylene and rehydrated in a graded ethanol series and endogenous peroxidase activity was inhibited by incubation with 3% $H_2O_2$. Sections were blocked with 10% goat serum in phosphate-buffered saline (PBS) and incubated overnight at 4 °C with primary antibodies. After a PBS wash, the sections were incubated with secondary antibody at 37 °C for 30 min. Immunohistochemical (IHC) staining was visualized by use of a diaminobenzidine kit (Zhongshan Goldenbridge Biotechnology, Beijing, China) according to the manufacturer's instructions. Sections were counterstained with hematoxylin to visualize nucleus. The primary antibodies included anti-CD123 Ab (1:100 dilution, ab21562, Abcam) and anti-KLF4 Ab (1:100 dilution, ab215036, Abcam).

Multiplex IHC with tyramide signal amplification (TSA) was performed using an Opal™ 4-Color Manual IHC Kit (AKOYA, NEL810001KT) according to the manufacturer's instructions. Briefly, sections of 4-μm thickness from human renal arteries were deparaffinized in xylene, rehydrated, and washed in tap water before boiling in AR buffer for epitope retrieval/microwave treatment (MWT). Protein blocking was performed using blocking buffer and incubating slides in a humidified chamber for 10 min at room temperature. Then, the slides were incubated with primary antibodies anti-CD123 (1:100 dilution, ab21562, Abcam), anti-SMα-actin (1:200 dilution, ab32575, Abcam), anti-CD68 (1:100 dilution, ab213363, Abcam), anti-KLF4 (1:100 dilution, ab215036, Abcam), and anti-PFKFB3 (1:100 dilution,

ab181861, Abcam) for 1 h at room temperature. Next, incubation with Polymer HRP Ms + Rb was performed at room temperature for 10 min. Opal signal was generated by incubating the slides with Opal Fluorophore Working Solution containing fluorophores DAPI, Opal 520, Opal 570, and Opal 690 at room temperature for 10 min. MWT was performed to strip the primary-secondary-HRP complex allowing introduction of the next primary antibody. For detection of the next target, restart the protocol at the blocking step. At last, the multiplex TSA stainings were enclosed in DAPI Fluoromount-G (SouthernBiotech, no. 0100-20). Visualization of 4-color Opal slides was performed using Olympus FV1200MPE microscope. Images were processed and quantified using Image J.

Immunofluorescence staining was performed with 4 μm paraffin cross-sections from the renal artery of human. After deparaffinized with xylene and rehydrated, the slides were pre-incubated with 10% goat serum and then incubated with primary antibodies anti-CD123 (1:100 dilution, ab21562, Abcam), anti-SMα-actin (1:200 dilution, ab32575 or ab240654, Abcam) and anti-LY6D (1:100 dilution, HPA024755, Merck). Secondary antibodies were fluorescein-labeled antibody to *rabbit* IgG (5230-0299, KPL, USA), fluorescein-labeled antibody to *mouse* IgG (5230-0307, KPL, USA), goat anti-*mouse* IgG H&L (Alexa Fluor® 647) (ab150115, Abcam) and goat anti-*rabbit* IgG H&L (Alexa Fluor® 647) (ab150079, Abcam).

Immunofluorescence staining of mouse aortic root paraffin sections was performed with primary antibodies anti-CD11c (1:100 dilution, ab11029, Abcam), anti-KLF4 (1:100 dilution, ab215036, Abcam), anti-PFKFB3 (1:100 dilution, ab181861, Abcam), anti-eEF1A2 (1:100 dilution, 16091-1-AP, Proteintech) and anti-SMα-actin (1:200 dilution, ab32575 or ab240654, Abcam). Secondary antibodies were fluorescein-labeled antibody to *rabbit* IgG (5230-0299, KPL, USA), fluorescein-labeled antibody to *mouse* IgG (5230-0307, KPL, USA), rhodamine-labeled antibody to *rabbit* IgG (5230-0332, KPL, USA), and rhodamine-labeled antibody to *mouse* IgG (5230-0336, KPL, USA). In each experiment, DAPI (157574, MB biomedical) was used for nuclear counterstaining. Images were captured by Olympus FV1200MPE microscope and processed and quantified using Image J.

HASMCs infected with pAd-GFP or pAd-GFP-KLF4 were fixed in 4% paraformaldehyde for 5 min at room temperature and then were washed with PBS, followed by incubation in 10% normal goat serum blocking solution for 30 min in a humidified chamber at room temperature. The cells were incubated with anti-CD123 (1:200 dilution, ab21562, Abcam) for 2 h at room temperature, washed with PBS, and incubated with rhodamine-conjugated secondary antibodies for 60 min. The cells were then washed with PBS, mounted with DAPI, and visualized with a laser scanning confocal microscope.

**Combined RNA in situ hybridization (ISH)/protein staining**. A biotin-labeled specific probe for circCTDP1 was designed and synthesized by GenePharma (Suzhou, China). The FISH experiment was performed according to the manufacturer's instructions (GenePharma). First, VSMCs were fixed in 4% paraformaldehyde for 15 min at room temperature and permeabilized using 0.2% Triton X-100. Then, cells were hybridized with specific probe of circCTDP1 overnight at 37 °C in hybridization buffer. For colocalization studies with eEF1A2, cells were fixed for 5 min in 2% formaldehyde after RNA ISH, and immunofluorescence was performed with anti-eEF1A2 (1:100 dilution, 16091-1-AP, Proteintech). Nuclei were counterstained with DAPI (SouthernBiotech, no. 0100-20). Cells were imaged using Olympus FV1200MPE microscope. The sequence of circCTDP1 probe used is as follows: GCACACACCAUCGCAGAGCGGUUCUGGUGAGGUU.

**In situ hybridization with proximity ligation assay (ISH-PLA)**. ISH-PLA was performed as previously described[5]. Briefly, *human MYH11* biotin-labeled probes were generated by Nick Translation (Roche, no. 10976776001) using biotin-14-dATP (Thermo Fisher, no. 19524016). Labeled probes (40 ng/slide) underwent denaturation in Hybridization Buffer (2×SSC, 50% high grade formamide, 10% dextran sulfate, 1 μg of *human* Cot-1 DNA) for 5 min at 80 °C. After immunostaining for SM α-actin (1:200 dilution, ab32575 or ab240654, Abcam), CD123 (1:100 dilution, ab21562, Abcam), and LY6D (1:100 dilution, HPA024755, Merck), slides were dehydrated in ethanol series and incubated in 1 mM EDTA (pH 8.0) for 20 min. Then, samples were incubated with pepsin (0.5%) in buffer (0.05 M Tris, 2 mM CaCl$_2$, 0.01 M EDTA, 0.01 M NaCl) at 37 °C for 20 min. Hybridization mixture containing biotin labeled probes was applied on sections. Sections were incubated at 80 °C for 5 min, followed by 24 h incubation at 37 °C. Hybridization was followed by multiple washes in 2×SSC, 0.1% NP-40 buffer. PLA was performed directly after ISH following the manufacturer's instructions (Merck, no. DUO92101). After blocking, sections were incubated with *mouse* H3K4dime (ab6000, Abcam) and *rabbit* Biotin (ab53494, Abcam) antibodies overnight at 4 °C, followed by incubation with secondary antibodies conjugated with PLA probe at 37 °C for 1 h. Then, ligation and amplification were performed. Finally, mounting medium with DAPI was used to coverslip the slides. Images were captured by Olympus FV1200MPE microscope. Settings were fixed at the beginning of both acquisition and analysis steps and were unchanged. Brightness and contrast were lightly and equally adjusted after merging. Images were processed and quantified using Image J.

**Western blot analysis**. Proteins from cultured HASMCs were prepared with lysis buffer (1% Triton X-100, 150 mM NaCl, 10 mM Tris-HCl, pH 7.4, 1 mM EDTA, 1 mM EGTA, pH 8.0, 0.2 mM $Na_3VO_4$, 0.2 mM phenylmethylsulfonyl fluoride, and 0.5% NP-40). Equal amounts of protein were separated on SDS-PAGE, and electrotransferred to a PVDF membrane (Millipore). Membranes were blocked with 5% milk in TTBS for 2 h at room temperature and incubated with primary antibodies overnight at 4 °C. Antibodies that were used are as follow: anti-PFKFB3 (1:1000, ab181861), anti-HK2 (1:1000, ab209847), anti-eEF1A2 (1:1000, 16091-1-AP), anti-KLF4 (1:1000, 11880-1-AP), anti-SMα-actin (1:1000, ab32575), anti-SM22α (1:1000, ab14106), anti-phospho-Stat3 (Tyr 705) (1:1000, Cell Signaling, catalog no. 9145), anti-phospho-Stat3 (Ser 727) (1:1000, Cell Signaling, catalog no. 94994), anti-acetyl-Stat3 (Lys 685) (1:1000, Cell Signaling, catalog no. 2523), anti-Stat3 (1:1000, Cell Signaling, catalog no. 9139), anti-phospho-Akt (Ser/Thr) (1:1000, Cell Signaling, catalog no. 9611), anti-Akt (1:1000, Cell Signaling, catalog no. 4691), anti-FLAG (1:2000, Sigma, catalog no. F365) or anti-β-actin (1:2000, sc-47778), membranes were then incubated with the HRP-conjugated secondary antibody (1:5000, Rockland) for 1 h at room temperature. The blots were treated with the Immobilon™ Western (Millipore), and detected by ECL (enhanced chemiluminescence) Fusion Fx (Vilber Lourmat).

**RNA preparation and quantitative real-time PCR**. Cultured HASMCs were lysed by using the QIAzol Lysis Reagent (Catalog no.79306). Total RNA was extracted from above sample according to the manufacturer's instructions (miRNeasy Mini Kit; Catalog no.217004). The quality of the RNA was determined using a Nanodrop 2000 (Thermo). cDNA was synthesized using an M-MLV First Strand Kit (Life Technologies) with random hexamer primers. Quantitative real-time PCR (qRT-PCR) of mRNAs or circRNAs was performed using Platinum SYBR Green qPCR Super Mix UDG Kit (Invitrogen) on a ABI 7500 FAST system (Life Technologies). Relative amount of transcripts was normalized with β-actin and calculated using the $2^{-\Delta\Delta Ct}$ formula. Supplementary Table 3 summarizes the primer sequences.

**Small interfering RNA and plasmid transfection**. Small interfering RNAs (siRNAs) targeting *human* PFKFB3 (si-PFKFB3 and si-PFKFB3#), eEF1A2 (si-eEF1A2 and si-eEF1A2#), KLF4 (si-KLF4 and si-KLF4#), circCTDP1 (si-circCTDP1 and si-circCTDP1#), circZFAT (si-circZFAT and si-circZFAT#), both CTDP1 linear and circular transcripts (si-both-C and si-both-C#), and both ZFAT linear and circular transcripts (si-both-Z and si-both-Z#) were designed by BioCaring Biotechnology (Shijiazhuang, China) and synthesized by GenePharma (Suzhou, China). The siRNA sequences were on Supplementary Table 4. Non-specific siRNA (si-Con), Akt siRNA (si-Akt), and Stat3 siRNA (si-Stat3) were purchased from Cell Signaling Technology. siRNAs and plasmid transfection were performed using Lipofectamine 2000 following the manufacturer's instructions. After transfection, HASMCs were treated with or without TMAO, pAd-eEF1A2, or pAd-GFP-KLF4. The medium was then collected for lactate production analysis, and cells were harvested and lysed for Western blotting or qRT-PCR.

**RNA immunoprecipitation (RIP) assay**. HASMCs were infected with pAd-GFP-KLF4 for 48 h, and then cells were used to conduct RIP experiments using primary antibodies eEF1A2 or IgG, and the Dynabeads™ Protein G Immunoprecipitation Kit (10007D, Thermo Fisher) according to the manufacturer's instructions. The RNA fraction isolated by RIP was quantified by NanoDrop 2000. The cDNA was synthesized using a M-MLV First Strand Kit (Life Technologies) with random hexamer primers. The RIP' circCTDP1 was subjected to qRT-PCR using the Platinum SYBR Green qPCR Super Mix UDG Kit (Invitrogen) and the ABI 7500 FAST system (Life Technologies).

**RNA pull-down assay**. HASMCs infected with pAd-GFP or pAd-GFP-KLF4 were washed in ice-cold PBS, lysed in 500 μl lysis buffer (20 mM Tris-HCl, pH 7.0, 150 mM NaCl, 0.5% NP-40, 5 mM EDTA, with freshly added 1 mM DTT, 1 mM PMSF, and 0.4 U/μl RNase inhibitor), and then incubated with 3 μg biotinylated DNA oligo probes against circCTDP1 back-splice sequence at 4 °C for 4 h. Then, the cells were incubated in 50 μl streptavidin-coated magnetic beads (SA1004; Invitrogen) at 4 °C for 2 h. RNase-free BSA and yeast tRNA (Sigma) were used to prevent the nonspecific binding of RNA and protein complexes. RNA bound to beads was extracted by TRIzol, while the bound protein was analyzed by Western blotting.

**CpG oligodeoxynucleotide (ODN) treatment and detection of interferon alpha (IFN-α) by ELISA**. HASMCs were infected with pAd-GFP or pAd-GFP-KLF4 for 24 h, or incubated with 200 mM TMAO for 24 h, and then treated with the adjuvant CpG ODN (6 μg/ml) for another 24 h. The endotoxin-free phosphorothioate-modified ODN was provided by GenePharma (small letters: phosphorothioate linkage; capital letters: phosphodiester linkage 3' of the base; bold: CpG-dinucleotides): ODN 2216: 5'-ggGGGA**CG**AT**CG**TCgggggG-3'; ODN 2243 (GC control to ODN 2216): 5'-ggGGGAGCATGCTCgggggG-3'. The culture medium was then collected and IFN-α production was detected using a *human* IFN-α ELISA kit (Proteintech, KE00044). The absorbance at 450 nm was measured with a microplate reader (SPECTRAFluor Plus, Tecan).

**Circulating TMAO quantification**. Mouse blood samples were collected into EDTA anticoagulant tubes after a 12 h overnight fast. The samples were immediately centrifuged at $3000 \times g$ for 15 min and plasma was stored at -80 °C and thawed fewer than three times. The TMAO level was quantified using stable isotope dilution liquid chromatography-tandem mass spectrometry (LC/MS/MS) in Peking University.

**Glycolytic activity assay**. The concentration of lactate in the cell medium was assessed with Lactate assay kit (ab65331, Abcam) following manufacturer's instructions. Briefly, fresh medium (ScienCell, no. 1101) containing 5.55 mM glucose were added to a 6-well plate cells, and lactate concentration in the medium was measured 1 h later and normalized to the number of cells in each well. Extracellular acidification rate (ECAR) was detected by Seahorse Bioscience XF-24 Extracellular Flux Analyzer. Briefly, HASMCs cultured in XF24-well cell culture microplates (Seahorse Bioscience) were infected with pAd-GFP or pAd-GFP-KLF4 for 48 h. ECAR was measured using the Seahorse XF glycolysis stress test Kit (Agilent) in XF base medium containing 10 mM glutamine but no glucose following sequential additions of glucose (10 mM), oligomycin (1 mM) and 2-DG (50 mM).

**Glucose consumption**. The glucose content in the medium was detected using a Glucose Assay kit (Merck, no. GAGO20) according to the manufacturer's protocols. The rate of glucose consumption was calculated according to the standard curve line and OD value of each sample. Then, the values were also normalized by cell numbers determined by using a blood cell counting chamber.

**Phosphoenol pyruvic acid (PEP) and TCA cycle intermediate measurements**. PEP and TCA cycle intermediates in HASMCs infected with pAd-GFP or pAd-GFP-KLF4 were determined by LC/MS/MS analysis. The chromatography system consisted of a Dionex Ultimate -3000 Reagent-FreeTM Ion Chromatograph (Dionex Corporation, Sunnyvale, CA, USA) with an AS auto sampler. The chromatographic system was coupled with an API 3200 Q TRAP instrument (AB SCIEX, USA) operated in a multiple reaction monitoring mode (MRM). Data were acquired and processed using Analyst v1.5.2 software (Applied Biosystems).

**Flow cytometry**. HASMCs infected with pAd-GFP or pAd-GFP-KLF4 were collected and washed with PBS and then resuspended in 100 μl FACS buffer. Fluorochrome-conjugated antibody for APC-CD123 (306012, BioLegend) was added for 30–45 min at room temperature. Isotype control antibody (400122, BioLegend) was added to samples at the same concentrations. After incubation, samples were washed three times and analyzed by FAC Sverse (FC500 MPL Beckman).

**mRNA microarray**. Total RNA was isolated with TRIzol reagent (Invitrogen) from HASMCs infected with or without pAd-GFP or pAd-GFP-KLF4 ($n = 3$ per group). The RNA quantity was assessed using a NanoDrop 2000, and RNA integrity was assessed by standard denaturing agarose gel electrophoresis. Microarray analysis was performed by Shanghai KangChen Biotech on an Agilent Array platform. The subsequent steps were performed according to the Agilent Whole Genome Oligo Microarray (one-color) protocol. Array data preprocessing, normalization, and quality control were conducted using GeneSpring software V12.1 (Agilent Technologies). To select differentially expressed genes, ratio change threshold values of g2.0 or e0.05 were used. Hierarchical clustering was performed using log2-transformed data in Cluster 3.0, and heat maps were generated with the Ggplot2 R environment. Gene Ontology (GO) term analysis was performed using the topGO R software. KEGG pathway analysis was applied to determine key signaling pathways and relationships between differentially expressed genes.

**circRNA microarray**. HASMCs were infected with or without pAd-GFP or pAd-GFP-KLF4 ($n = 3$ per group). Then the RNA isolation and microarray analysis of *human* circRNAs were performed by KangChen BioTech (Shanghai). Total RNAs were isolated and digested with RNase R (Epicentre, Inc.) to remove linear RNAs. The enriched circRNAs were amplified and transcribed into fluorescent cDNA utilizing a random priming method and hybridized onto the Arraystar *Human* circRNA Array V2. CircRNAs were selected based on being significantly differentially expressed (fold changes ≥ 1.5 and p values ≤ 0.05).

**TMT-based LC-MS/MS analysis**. HASMCs were infected with pAd-GFP, pAd-GFP-KLF4, or pAd-eEF1A2. Then the protein isolation and TMT-based LC-MS/MS analysis were performed by KangChen BioTech (Shanghai). The cells were lysed with RIPA lysis buffer, and the protein was then isolated. The protein content was determined by BCA assay. 100 μg protein for each sample was used for TMT labeling and LC-MS/MS in the following steps: (1) reduction and alkylation; (2) acetone precipitation; (3) re-suspend protein for tryptic digest; (4) TMT labeling; (5) cleaning up of SDC; (6) peptide desalting for LC-MS/MS; and (7) LC-MS/MS: 100 μg peptides were fractionated to 120 fractions with high pH RPRP-HPLC, and then combined to 8 fractions. For each fraction, 2 μg peptides were separated and analyzed with a Nano-HPLC (EASY-nLC1200) coupled to Q-Exactive mass

spectrometry (Thermo Finnigan). Proteins were selected based on being significantly differentially expressed (fold changes ≥1.2).

**Statistics and reproducibility**. All of the data are presented as the means ± SEM. Differences between two groups were assessed using analysis of variance followed by a Student's $t$ test. For multiple comparisons, one-way ANOVA followed by Tukey's multiple comparison tests was used. A value of $P < 0.05$ was considered statistically significant, and denoted with 1, 2, 3, or 4 asterisks when lower than 0.05, 0.01, 0.005, or 0.001, respectively. Information on the reproducibility of experiments, including the sample sizes and number of replicates and how replicates were defined in the legends of figures.

**Reporting summary**. Further information on research design is available in the Nature Portfolio Reporting Summary linked to this article.

## Data availability

Uncropped western blot images are shown in Supplementary Figs. 13–20, and all source data underlying the graphs in the main figures are shown in Supplementary Data. The mRNA and circRNA microarray data have been deposited in the Gene Expression Omnibus database (GSE216303 and GSE216305). Newly generated plasmids in this study are deposited in Addgene (https://www.addgene.org/) with the deposition ID numbers 193961 and 193957. The TMT-based LC-MS/MS data have been deposited in the iProX database (https://www.iprox.cn/page/home.html) with the deposition ID number IPX0005376000. All other datasets generated and/or analyzed during this study are available from the corresponding author on reasonable request.

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

## Acknowledgements

We thank Dr. Zhan Yang for providing the patent of circRNA overexpression vector (Patent No. ZL201710215532.0). We thank Xuerui Fu from Hebei Medical University Core Facilities and Centers for confocal fluorescent measurements. This work was supported by the National Natural Science Foundation of China (No. 81770285, No. 31671182, No. 31871152, No. 81971328) and the Natural Science Foundation of Hebei Province of China (No. H2022206074, No. H2021206459).

## Author contributions

X.Z., B.Z., L. Zhao, and J.S. designed and performed most experiments and data analysis. Z.Y., Y.Z., R.F., L. Zheng, M. Zhao, and H.L. conducted some experiments. M. Zhang and D.M. collected human tissue. J.W. supervised the work and wrote the manuscript.

## Competing interests

The authors declare no competing interests.
