## [Peer Review File · Communications Biology]

Reviewers' comments:

Reviewer #1 (Remarks to the Author):

In the present manuscript, Zhang and colleagues investigated the role of the axis KLF4-PFKFB3 in the phenotypic switch of VSMCs. Via an integrated approach, the authors tried to highlight a very complex pathway in which a transcription factor, a translation modulator, a circular RNA, and a metabolic enzyme are involved. This is a huge effort, however, due to the complexity of the analysis, the paper falls short in different aspects, that must be further investigated. Below a specific review for the authors:

1. The authors stated that KLF4 triggers a switch of VSMCs toward a pCD-like phenotype. In my opinion, the authors must provide further data showing that this is happening. First, in the analyzed plaques (figure 1), there is no co-staining KLF4-CD68-ACTA2, this would really suggest that such transition might happen. Then, in general, all immunofluorescence data in the paper must be accompanied by quantification and statistical analysis using different samples, with sufficient power analysis.
2. About the KLF4-GFP construct used for the gain-of-function experiments, is it a bicistronic vector of it generates a fusion KLF4-GFP? This is important because when it expresses KLF4 the GFP localization is mainly nuclear.
3. The conclusion reached with the experiments reported in the first paragraph is not totally supported, because, although the modulation of gene expression might suggest a switch due to KLF4, no metabolic/biological variations are observed (IFN production). Therefore, all structure of the work is limited by that and not fully supported, thus these functional data are essential.
4. The functional data about the modulation of glycolysis in KLF4-overpressing cells are interesting, but does the contrary happen in loss-of-function conditions? If so, will the authors be able to perform a rescue experiment with cells transduced with siRNAs vs KLF4 and a construct expressing a not targetable KLF4 of a different origin, such as mouse one?
5. Is there, in KLF4-expressing VSMCs, an alteration of the expression of fundamental glycolytic genes, such as HKII?
6. In the WB for eEF1A2 (figure 4) there are two bands, but only one is reduced in presence of specific siRNAs, what is the other one?
7. The link between the different studied molecular pathways and circRNAs is a very long shot, what prompted the authors to study that?
8. Did the authors experimentally validate the identified circRNAs? I see only qPCR data, and while circCTDP1 was already reported and validate by others, none is available on circZFAT. There are pipelines of wet experiments that will help to demonstrate whether potential circRNAs are really circular.
9. WB in figure 5 must be quantified and statistical analysis on different samples reported. This is a general comment for all experiments.
10. If circCTDP1 influences PFKFB3 translation, it should be localized at the ER. The authors should show the physical co-localization of circCTDP1 and eEF1A2 in cells by a combined in situ/protein staining. There are several reports showing the potentiality of this approach.
11. The authors must also measure the protein level of PFKFB3 in cells treated with 2-DG and PFK15.
12. I did not understand the conclusion drawn by the authors at page 8 line 280: Did they do the same experiments also in other cell types?
13. For the TMAO experiments, the conclusion at page 8 line 307 is not supported by experimental evidence. The authors must show that TMAO alters the biology of cells with specific functional assays.
14. Then, when using siRNAs vs KLF4, the results must be validated with at least two different oligos, in order to be sure that the observed phenotype is not due to off-targets. For these experiments, it is clear that the authors used only one siRNA (the best among the two tested), thus the same comment refers also to all reported experiments. As an alternative to using two different siRNAs, the author can also perform a rescue experiment in which the silenced gene is re-expressed using a species analog (i.e. human vs mouse). However, one of the two proposed approaches has to be done.
15. Is circCTDP1 expression influenced by PDGF-BB?

16. Data on Apoe ko mice must be better presented. Usually, to have consistency among different animals, the tissue sections must be obtained close to the aortic valve, where usually plaque form. Furthermore, to draw conclusions, a statistical quantification must be also enclosed together with the representative images.

17. Data presented in figure 8d must be completed with the labeling of CD123.

18. We strongly suggest thoroughly proofread the manuscript to improve the overall quality of the scientific communication.

Minor points:

1. Data showed in Figures 8a, b and c can be moved to the supplemental section.

2. Panel g figure 6, there is no statistical indication for siSTAT3.

Reviewer #2 (Remarks to the Author):

Interesting manuscript with well-performed experiments and solid data. However, a few remarks need to be addressed:

- While the manuscript is in general well-written, the coherence between sentences, in particular the introduction, is lacking. Especially between the different paragraphs in the introduction addressing all the different previous findings (circRNAs, TMAO, metabolism, KLF4, CD123). This makes the introduction hard to read. While in the result resection the structure appears to be there. Please rewrite the introduction, since placing the information into context would increase the readability of the manuscript.

- Why did the authors focus on CD123? Please introduce this as well.

- Please provide the quantification of the pixel overlap between the markers used for confocal imaging analysis (eg. Fig 1C and S1).

- Could the authors explain the partial expression of CD123 in the nucleus in Fig 1G?

- Does reduction in KLF4 expression (siRNA) reduce CD123 expression?

- Please include glucose levels in medium as well, besides measuring lactate levels to solidify the glycolysis data.

- The authors should include their main findings in the histological plaques as well (e.g confocal microscopy and stain for PFKFB3 expression in combination with KLF4))

- To further test whether VSMC-derived pDC-like cells were able to produce interferon alpha (IFN- α), which is an important function of matured pDCs, we treated KLF4-overexpressing VSMCs with CpG oligodeoxynucleotides (ODN), an inducer of production of IFN- α , and detected IFN- α production. This sentence needs to be rewritten, since it's stated at the end that IFN α production was detected, it insinuates there is a difference between ctrl and KLF4-OE cells.

- Does a knock-down of PFKFB3 in KLF4 over-expressing cells affect CD123 expression? This question came up when reading the first PFKFB3 knock-down experiments, however this question is answered in Figure 6. Please move this data up.

Reviewer #3 (Remarks to the Author):

What are the major claims of the paper? Are they novel and will they be of interest to others in the community and the wider field? If the conclusions are not original, it would be helpful if you could provide relevant references.

Atherosclerosis is a major underlying cause of cardiovascular related death and understanding how to ameliorate or prevent adverse complications from atherosclerosis, including plaque rupture or erosion, is a major focus of preclinical and clinical research. Historically, much of this focus has been on

reducing cholesterol, leading to the widespread use of statins, and on detrimental inflammation. Recently, smooth muscle cells have become a target to increase plaque stabilization, and in the process, metabolic state and regulation has emerged as a major player in smooth muscle cell phenotypic switching and ECM deposition. The authors expand on previous work detailing the importance of metabolic state in atherosclerosis, and in SMC in particular. In this manuscript, the authors provide evidence that SMC to pDC phenotypic switching is regulated by KLF4, PFKFB3, eEF1A2, and circCTFP1.

While the authors clearly layout the role KLF4 plays in regulating SMC metabolic state in vitro, this was first shown, although not in as much detail, in previous work by Alencar et al (Circ 2020) where knockout of KLF4 in SMC showed significant changes in metabolic pathways by bulk RNAseq, as well as resulting in SMC modulation based on immunofluorescence and scRNAseq analyses. Other relevant papers in the field of atherosclerosis, metabolism, and SMC include: Newman et al Nature Metab 2021 (SMC phenotypic transitions and ECM synthesis are regulated by metabolic state), Tomas et al EHJ 2018 (metabolism in symptomatic vs non-symptomatic lesion), and the multiple papers from the 1980s including Lynch and Paul Experientia 1985 (energy metabolism and contractility ability in SMC). The novelty of this manuscript stems mostly from the identification of the small non coding RNA effectors of phenotypic switching, especially as it relates to post-RNA or post-translational modification of protein as a way to induce phenotypic changes via modulating metabolic state. This is quite exciting and this idea in particular will be important for the eventual goal of targeting individual cell types in disease. However, the conclusions about phenotypic state as well as impact on atherosclerosis in this paper are not entirely supported by the data. Nevertheless, identification of this RNA modifier axis has the potential to influence thinking with substantial clarifications and/or revisions.

Is the work convincing, and if not, what further evidence would be required to strengthen the conclusions?

The authors outline putative SMC to pDC phenotypic switching and correlate that to atherosclerotic lesion progression. The SMC to pDC switch itself is, to this reviewer's knowledge, unknown previously. However, the identification of SMC-derived pDCs is solely reliant on marker protein co-localization of SMC marker proteins and pDC marker proteins. This is problematic for a number of reasons including the ubiquity of many of the markers used by the authors in atherosclerotic lesions, the propensity of SMC themselves to downregulate their own marker proteins (rendering use of ACTA2 in vivo unreliable), and the relative inabundance of pDCs in atherosclerotic lesions (as cited in ref 32). Since this is a main point the authors are trying to make, it is imperative that evidence for this be incontrovertible.

Major comments:

1. SMC to pDC phenotypic switching. To this reviewer, the almost complete co-localization of ACTA2 staining with CD123 (pDC marker) is suspect as authors seem to imply that all ACTA2+ (putative SMC) have transitioned to a pDC phenotype. Couple this with the multiple SMC scRNAseq studies over the past few years (Wirka et al, Pan et al, Alencar et al) that do not show a pDC-like cluster derived from murine sorted lineage tagged SMC, this reviewer is not convinced of the author's claims of SMC to pDC transition in atherosclerosis. Further, there is no attempt to positively identify the ACTA2+ cells as SMC. This reviewer suggests confirming SMC origin by more than ACTA2 expression as well as providing more substantial evidence than marker staining that pDC-like cells exist, as this is the crux of the author's story. While the reviewer is aware that lineage tracing is impossible in humans, claims of SMC-derived pDC can be supported by either In Situ Hybridization with Proximity Ligation Assay (Gomez et al Nat Meth 2012) and/or use of the many murine SMC-lineage tracing models available.
2. pDC specificity marker CD123. The idea of SMC can differentiate to a pDC-like cell is bolstered by the in vitro work showing increased CD123 and other pan DC mRNA expression (sometimes 3000-fold!) after KLF4 OE, however, authors do not evaluate any other marker of pDC (except for pan CD11c in mice) in vivo nor functionality aside from increased IFN α expression. Considering CD123 in disease is also used as a marker for hematopoietic stem cells, the reviewer suggests including

additional metrics to determine a pDC-like state in vivo. Authors may consider including data using KLF4 KD vs. wild type in order to understand if this occurs normally, or is a function of the OE.

3. IF staining controls. The reviewer questions the settings used for analysis of co-localization in Figures 1 and 8 and Supp Figs 1 and 5. The images presented look over-processed and/or over-blown. This especially apparent in the media and the body of the lesion where there is an unexpected morphology of the ACTA2 staining as well as the seemingly 100% co-localization of multiple markers including CD68, CD123, and ACTA2. Most previous reports, including ones the author cites (reference 4) show that about 10% of the lesion cells are ACTA2+ and that only a very small fraction of ACTA2+ cells co-stain for macrophage markers. The reviewer urges the authors to show the unprocessed images and to provide images of negative controls and more detailed information about assessment of co-localization and relative abundance of cell types in the methods.

4. Quantification of claims:

a. Please quantify the percentage of ACTA2 cells that co-localize with the various markers in the figures (e.g. what is one to understand from "...obvious reduction in the expression of these genes"? page 9 line 345). Please also quantify plaque metrics to bolster claims of "enhance[d] atherosclerosis".

b. The reviewer suggests including a description of and relative abundance of pDC in plaques. The authors suggest that an increase in ACTA2+ CD123+ cells in atherosclerotic lesions in mice is associated with increased plaque size (without quantification), but aside from this, there is no clear stated directionality on if these metabolic changes are beneficial or detrimental, nor if changes (purportedly increases) in pDC-like cells are protective or pro-atherogenic (excluding plaque size, which is not the best indicator of stability, for references see the works of Virmani et al from CVPath).

5. Conclusions. Because of the lack of lineage tracing, quantification of lesion area, cell phenotype, and negative controls for immunofluorescence images, the authors are suggested to tone down a number of the major conclusions in the paper.

a. On Page 10 paragraph 1, the authors suggest that ACTA2 staining is indicative of a SMC origin, but do not take into account the fact that in mice, nearly 80% of the SMC-derived cells have lost their ACTA2+ marker expression, especially those in the lesion core (ref 4).

b. In SFig 1, authors note that "pDC marker CD123 and macrophage marker CD68 are abundantly expressed in VSMCs..." indicating that CD123 and CD68 are co-expressed, but the idea that the pDC are derived from monocytes/macrophages is not explored. Indeed, monocytes and pDC share a known common progenitor (HSC, MPP), whereas SMC and pDC do not share a known progenitor, is it possible that 1. The pDC in the lesion are derived from the CD68 cells (i.e. myeloid cells that may have acquired ACTA2 expression, Newman Nat Metab 2021, Albarran-Juarez Atherosclerosis 2016, Caplice PNAS 2003) or 2. the co-localization of these three markers in the plaque is an artifact of staining?

c. The role of KLF4 in SMC phenotypic switching is well-known, however the authors suggest that KLF4 exclusively regulates the SMC to pDC switch. Indeed, Alencar et al (Circ 2020) show a number of KLF4-regulated SMC-derived phenotypes in atherosclerosis including a chondrocyte-like cell and other osteogenic phenotypes but not a pDC-like phenotype. Nor is this shown in the other related scRNAseq studies of SMC-labeled cells in atherosclerosis (Wirka et al, Pan et al). Conclusions would be supported if authors show this unique KLF4-PFKFB3-RNA axis is specific for pDC transitions.

Minor comments:

1. Please define PFKFB3 page 4 line 107

2. Figure 1c, SFig 1a, SFig 5: please indicate if the field of view is in the core or the cap region of the lesion

3. Please add graphs quantifying the overlap of ACTA2 cells that co-localize with the various markers in the figures (i.e. what is one to understand from "...obvious reduction in the expression of these genes"? page 9 line 345)

4. Please add detail to the methods about cell culture conditions including duration of treatments

5. Please ensure conclusions are borne out in the data or clarify/tone down conclusions not directly show in the manuscript. For example: clarify claim on pg 11 line 429 that the "...glycolytic shift...can also activate Akt" by showing these metabolic using Seahorse with glycolysis agonists and KLF4 OE (refer to page 8 line 286, where it "evidently" causes changes).

6. Regarding Akt expression in atherosclerosis pathogenesis, authors show Akt is phosphorylated after

KLF4 OE as part of the PFKFB3 axis and later suggest that this accelerates atherosclerosis pathology. A seminal paper in the field (Fernandez-Hernando ATVB 2009) shows Akt1 is required for SMC survival and for overall plaque stabilization. How do the authors claims fit in with the wider field?

7. The authors should include in the figure legends statistical tests performed, error bars (SEM or SD?), replicate numbers, and scale bars for all images to facilitate assessment of statistical analysis and validity of data.

Point-by-point response to reviewer 1

In the present manuscript, Zhang and colleagues investigated the role of the axis KLF4-PFKFB3 in the phenotypic switch of VSMCs. Via an integrated approach, the authors tried to highlight a very complex pathway in which a transcription factor, a translation modulator, a circular RNA, and a metabolic enzyme are involved. This is an effort, however, due to the complexity of the analysis, the paper falls short in different aspects, that must be further investigated. Below a specific review for the authors:

Q1: The authors stated that KLF4 triggers a switch of VSMCs toward a pCD-like phenotype. In my opinion, the authors must provide further data showing that this is happening. First, in the analyzed plaques (figure 1), there is no co-staining KLF4-CD68-ACTA2, this would really suggest that such transition might happen. Then, in general, all immunofluorescence data in the paper must be accompanied by quantification and statistical analysis using different samples, with sufficient power analysis.

A1: Thank you for this helpful suggestion. We performed the co-staining experiments for KLF4/CD123/ACTA2 and KLF4/CD68/ACTA2 using a multiplex tyramide signal amplification (TSA) staining, which is open and flexible for use with any primary antibody without cross-reactivity. The results showed that pDC marker CD123 and macrophage marker CD68 were colocalized with VSMCs in the analyzed plaques (Please refer to Figs. 1c, d and Supplementary Fig. 1). Experimental methods and results were described, respectively, in Method and Result section. As a result, we observed 19% and 12% of SM α -actin positive (SMA- α^+) cells that expressed CD123 and CD68, respectively, in human atherosclerotic lesions. Importantly, all the CD123⁺SMA- α^+ and CD68⁺SMA- α^+ cells were KLF4 positive. All immunofluorescence data have been quantified and statistically analyzed accordingly.

Q2: About the KLF4-GFP construct used for the gain-of-function experiments, is it a bicistronic vector or it generates a fusion KLF4-GFP? This is important because when it expresses KLF4 the GFP localization is mainly nuclear.

A2: The KLF4-GFP construct used for the gain-of-function experiments generates a fusion KLF4-GFP. GFP was fused to the C-terminal of KLF4 cDNA, the termination codon of KLF4 was removed, and then the fusion fragment was cloned into adeno-associated virus plasmid for adeno-associated virus packaging, with the promoter as CMV.

Q3: The conclusion reached with the experiments reported in the first paragraph is not totally supported, because, although the modulation of gene expression might suggest a switch due to KLF4, no metabolic/biological variations are observed (IFN production). Therefore, all structure of the work is limited by that and not fully supported, thus these functional data are essential.

A3: We thank the reviewer's constructive comment. Because producing abundant

interferon alpha (IFN- α) upon stimulation by foreign nucleic acids is a unique capacity of pDCs, we tested whether KLF4-overexpressing cells treated with CpG oligodeoxynucleotides (ODN) were able to produce IFN- α in the previous manuscript. Results showed that IFN- α -producing activities were hardly detectable in the KLF4-overexpressing VSMCs, and thus they remain distinguishable from the authentic pDCs. Consistently, a subset of studies showed that SMC-derived cells within atherosclerotic lesions express multiple markers of other cells, such as macrophages (M ϕ s) and mesenchymal stem cells (MSCs), but they did not appear to function as authentic M ϕ s or MSCs (Owens et al, Nat Med, 21(6):628-37; Fisher et al, Arterioscler Thromb Vasc Biol, 35(3):535-46). What the function of the phenotypically modulated VSMCs has and how their functional properties affect atherogenesis are of great challenge and need further studies in the future.

As raised in the reviewer comments, although the modulation of gene expression might suggest a switch due to KLF4, no metabolic/biological variations are observed in this study. Considering the fact that this study indeed demonstrated a phenotypic switch of KLF4-overexpressing VSMCs including down-regulation of classical VSMCs markers and acquisition of pDCs-associated features in vitro and in vivo and the fact that KLF4-overexpressing cells indeed lack the main functional properties of pDCs, we revised the main conclusion in the first paragraph as “KLF4 plays a critical role in triggering pDC marker expression in VSMCs” (**page 4 line 112**) and toned down some conclusive statements. For example, the two sentences “These findings suggest that VSMCs within the plaques switch to a pDC-like phenotype and that KLF4 upregulation is correlated with VSMC phenotypic transitions” and “Taken together, these results indicate that KLF4 upregulation is responsible for the switching of VSMCs to pDC-like phenotype” in the previous manuscript were summarized into one sentence “These findings suggest that VSMCs within the plaques may switch to a pDC-like phenotype and that KLF4 upregulation is correlated with this phenotypic change” (**page 4 line 133**).

Q4: The functional data about the modulation of glycolysis in KLF4-overexpressing cells are interesting, but does the contrary happen in loss-of-function conditions? If so, will the authors be able to perform a rescue experiment with cells transduced with siRNAs vs KLF4 and a construct expressing a not targetable KLF4 of a different origin, such as mouse one?

A4: We have performed the loss-of-function experiments with siRNA vs KLF4, however, knockdown of KLF4 showed little effect on lactate production and PFKFB3 expression (data not shown and Fig. 7o), although silencing KLF4 suppressed TMAO-induced expression of PFKFB3 (Fig. 7o). A possible reason is that KLF4 overexpression might reach supra-physiological levels. However, the dose of adenovirus had been titrated and tracked relative to endogenous KLF4 levels before using. On the other hand, the

specificity of gene expression modulated by KLF4 overexpression was confirmed by the fact that KLF5 overexpression did not affect these gene expressions (Figs. 1f and 4g).

Q5: Is there, in KLF4-expressing VSMCs, an alteration of the expression of fundamental glycolytic genes, such as HKII?

A5: Western blot analysis was performed to detect the expression of HK2 in KLF4-overexpressing VSMCs. Consistent with the mRNA microarray analyses of KLF4-overexpressing VSMCs and the TMT-based LC-MS/MS data (Fig. 3a, b), overexpression of KLF4 in VSMCs did not affect HK2 expression (Fig. 3d, e).

Q6: In the WB for eEF1A2 (figure 4) there are two bands, but only one is reduced in presence of specific siRNAs, what is the other one?

A6: As the reviewer raised, we also observed there are two bands in the WB for eEF1A2 when performed experiments. We analyzed that the band with lower molecular weight is nonspecific with nearly the same quantity in each group. It does not seem to be the other subunit of eEF1A, eEF1A1, because the molecular weight of the two isoforms is almost the same. Importantly, the efficacy and specificity of siRNAs had been also confirmed by qRT-PCR, showing a significant reduction of eEF1A2 mRNA level (75%) but not the eEF1A1 mRNA (data not shown).

Q7: The link between the different studied molecular pathways and circRNAs is a very long shot, what prompted the authors to study that?

A7: We and others observed that circRNAs play fundamental roles in cardiovascular diseases through their miRNA/protein-binding capacity (Circ Res, 2017,121:628-635; Nucleic Acids Res, 2019,47:3580-3593; Theranostics, 2017,7:3842-3855). Thus, we focused the coactivator for eEF1A2 on circRNAs and performed circRNA microarrays to screen differentially expressed circRNAs regulated by KLF4 and established a method to construct circRNA overexpression vector (Patent No. ZL201710215532.0).

Q8: Did the authors experimentally validate the identified circRNAs? I see only qPCR data, and while circCTDP1 was already reported and validate by others, none is available on circZFAT. There are pipelines of wet experiments that will help to demonstrate whether potential circRNAs are really circular.

A8: The presence of circZFAT and circCTDP1 had been validated by using divergent primers to amplify circRNAs formed by head-to-tail splicing. We added the data to the Supplementary figure section in the revised version (Supplementary Fig. 7).

Q9: WB in figure 5 must be quantified and statistical analysis on different samples reported. This is a general comment for all experiments.

A9: All the WB in this study has been quantified in the revised version, and statistical analysis was performed on different samples.

Q10: If circCTDP1 influences PFKFB3 translation, it should be localized at the ER. The authors should show the physical co-localization of circCTDP1 and eEF1A2 in cells by a

combined in situ/protein staining. There are several reports showing the potentiality of this approach.

A10: The physical co-localization of circCTDP1 and eEF1A2 in cells has been confirmed by a combined in situ/protein staining in the revised version (Fig. 5j).

Q11: The authors must also measure the protein level of PFKFB3 in cells treated with 2-DG and PFK15.

A11: When the activities of glycolytic enzymes, hexokinase and PFKFB3, were inhibited by 2-DG and PFK15, respectively, PFKFB3 expression was not altered as demonstrated by Western blot analysis in the revised version (Supplementary Fig. 10a).

Q12: I did not understand the conclusion drawn by the authors at page 8 line 280: Did they do the same experiments also in other cell types?

A12: Takashi Iwamoto et al showed that STAT3 activation (Tyr705 phosphorylation) is essential for CD123 expression in mouse leukemia cells (Cytokine, 2004, 25: 136-139), as described in the manuscript. However, in VSMCs, we observed that inhibition of Akt activation but not STAT3 blocked the upregulation of CD123 expression induced by KLF4. Thus, we drew the conclusion that “This implies that the effect of STAT3 activation on CD123 expression is cell-context dependent”.

Q13: For the TMAO experiments, the conclusion at page 8 line 307 is not supported by experimental evidence. The authors must show that TMAO alters the biology of cells with specific functional assays.

A13: In the revised version, we supplemented the experiment to observe the effect of TMAO on IFN- α production in VSMCs. As expected, TMAO treatment could not induce IFN- α production (Supplementary Fig. 11c), although it could induce CD123 expression and inhibited SM22 α and SM α -actin expression (Figs. 7c, g and Supplementary Fig. 11a, b). Just as we mentioned in response to Q3, we toned down this conclusive statement and revised it as “These results indicated that TMAO could convert VSMCs to a dysfunctional pDC-like cell” (**page 8 line 294 and page 8 line 308**).

Q14: Then, when using siRNAs vs KLF4, the results must be validated with at least two different oligos, in order to be sure that the observed phenotype is not due to off-targets. For these experiments, it is clear that the authors used only one siRNA (the best among the two tested), thus the same comment refers also to all reported experiments. As an alternative to using two different siRNAs, the author can also perform a rescue experiment in which the silenced gene is re-expressed using a species analog (i.e. human vs mouse). However, one of the two proposed approaches has to be done.

A14: Thank you for your good suggestion. For all of the loss-of-function experiments with siRNAs, we used a different siRNA to repeat the experiments and got the same results (Supplementary Figs. 3, 4b, 8, 10b, and 11e-i).

Q15: Is circCTDP1 expression influenced by PDGF-BB?

A15: circCTDP1 expression was not influenced by PDGF-BB as demonstrated by quantitative RT-PCR (Fig. 7t).

Q16: Data on Apoe ko mice must be better presented. Usually, to have consistency among different animals, the tissue sections must be obtained close to the aortic valve, where usually plaque form. Furthermore, to draw conclusions, a statistical quantification must be also enclosed together with the representative images.

A16: As the reviewer suggested, mouse aortic root paraffin sections were stained using immunofluorescence staining and the immunofluorescence data have been quantified and statistically analyzed (Figs. 8 and Supplementary Fig. 12d-f).

Q17: Data presented in figure 8d must be completed with the labeling of CD123.

A17: CD123 is a marker known to be expressed on human pDCs but not mouse DCs, whereas CD11c is mainly expressed on mouse DCs and pDCs, thus, we stained CD11c using the mouse vessel.

Q18: We strongly suggest thoroughly proofread the manuscript to improve the overall quality of the scientific communication.

A18: The manuscript has been proofread by a native English speaker familiar with this type of research.

Minor points:

Q1: Data showed in Figures 8a, b and c can be moved to the supplemental section.

A1: Data shown in Fig. 8a-c in the previous manuscript have been moved to the Supplemental Figure section (Supplementary Fig. 12a-c) in the revised version.

Q2: Panel g figure 6, there is no statistical indication for siSTAT3.

The statistical indication for siSTAT3 in Fig. 6g in the previous manuscript has been added in the revised version (Fig. 6f).

Point-by-point response to reviewer 2

Reviewer #2 (Remarks to the Author):

Interesting manuscript with well-performed experiments and solid data. However, a few remarks need to be addressed:

Q- While the manuscript is in general well-written, the coherence between sentences, in particular the introduction, is lacking. Especially between the different paragraphs in the introduction addressing all the different previous findings (circRNAs, TMAO, metabolism, KLF4, CD123). This makes the introduction hard to read. While in the result resection the structure appears to be there. Please rewrite the introduction, since placing the information into context would increase the readability of the manuscript.

A: As the reviewer suggested, we rewrote the introduction and made it more coherent and

concise. It should be noted that we introduced PFKFB3 in the Introduction section in the revised version according to reviewer 3's comments.

Q- Why did the authors focus on CD123? Please introduce this as well.

A: Because in addition to macrophages (Mφs) and mesenchymal stem cells (MSCs) markers, VSMCs in human atherosclerotic lesions also express CD123, a pDC marker, and CD123-positive pDCs have been previously detected in atherosclerotic plaques by electron microscopy and immunohistochemistry (Grassia et al, Pharmacol Ther, 2013, 137:172-182; Daissormont et al, Circ Res, 2011, 109:1387-1395). Thus, it is likely that VSMCs also have the potential to switch to DCs. However, little is known about this switch and the role that KLF4 plays in this switch during atherogenesis. Thus, we focused on CD123 and KLF4 regulation of CD123 expression. As the reviewer suggested, we introduced this in the Introduction section in the revised version (**page 3 line 64**).

Q- Please provide the quantification of the pixel overlap between the markers used for confocal imaging analysis (eg. Fig 1C and S1).

A: According to the reviewer 1's comment to show the co-staining of KLF4-CD123/CD68-ACTA2 in atherosclerosis lesion, we improved the experimental method and performed a multiplex tyramide signal amplification (TSA) staining, which is open and flexible for use with any primary antibody without cross-reactivity. The images were shown in Figs. 1c, d and Supplementary Fig. 1. The staining data have been quantified and statistically analyzed accordingly.

Also, according to the reviewer 1's comment to present data on ApoE ko mice with the tissue sections obtained close to the aortic valve (data was from the carotid artery of mouse in the previous manuscript), we performed immunofluorescence staining using mouse aortic root paraffin sections and the immunofluorescence data have been quantified and statistically analyzed accordingly (Figs. 8 and Supplementary Fig. 12d-f).

Q- Could the authors explain the partial expression of CD123 in the nucleus in Fig 1G?

A: We considered that it might be a result of the high expression of CD123 induced by KLF4 overexpression and supra-physiological levels of CD123 result in its partial nuclear distribution.

Q- Does reduction in KLF4 expression (siRNA) reduce CD123 expression?

A: We had performed the loss-of-function experiments with siRNA vs KLF4, however, knockdown of KLF4 showed little effect on CD123 expression. A possible reason is that KLF4 overexpression might reach supra-physiological levels. However, the dose of adenovirus had been titrated and tracked relative to endogenous KLF4 levels before using. On the other hand, the specificity of gene expression modulated by KLF4 overexpression was confirmed by the fact that KLF5 overexpression did not affect these gene expressions (Figs. 1f and 4g). We reasoned that CD123 expression in control VSMCs is extremely low and almost undetectable (Figure 1). Thus, its expression was easily induced but could not

be reduced experimentally.

Q- Please include glucose levels in medium as well, besides measuring lactate levels to solidify the glycolysis data.

A: Thank you for this good suggestion. We measured the glucose levels in medium (Fig. 2b) in the revised version. Experimental method and result were described in Method and Result section, respectively.

Q- The authors should include their main findings in the histological plaques as well (e.g confocal microscopy and stain for PFKFB3 expression in combination with KLF4))

A: We thank the reviewer's constructive comment. The co-staining for KLF4/PFKFB3/SMA- α was performed in human atherosclerotic plaques using a multiplex tyramide signal amplification (TSA) staining (Figure 3l). As a result, PFKFB3 was mainly localized in the nucleus, consistent with the previous report (Li et al, Nat Commun, 2018;9:508) and largely co-localized with KLF4 in the atherosclerotic lesion.

Q- To further test whether VSMC-derived pDC-like cells were able to produce interferon alpha (IFN- α), which is an important function of matured pDCs, we treated KLF4-overexpressing VSMCs with CpG oligodeoxynucleotides (ODN), an inducer of production of IFN- α , and detected IFN- α production

This sentence needs to be rewritten, since it's stated at the end that IFN α production was detected, it insinuates there is a difference between ctrl and KLF4-OE cells.

A: This sentence has been rewritten as "To further test the ability of VSMC-derived pDC-like cells to produce interferon alpha (IFN- α), which is an important function of matured pDCs, we treated KLF4-overexpressing VSMCs with CpG oligodeoxynucleotides (ODN), an inducer of production of IFN- α , and measured the IFN- α levels in medium" in the revised version (page 4 line 135).

Q- Does a knock-down of PFKFB3 in KLF4 over-expressing cells affect CD123 expression? This question came up when reading the first PFKFB3 knock-down experiments, however this question is answered in Figure 6. Please move this data up.

A: Thanks for your careful review. We have moved this data from Figure 6 to Figure 3k in the revised version.

Point-by-point response to reviewer 3

Reviewer #3 (Remarks to the Author):

What are the major claims of the paper? Are they novel and will they be of interest to others in the community and the wider field? If the conclusions are not original, it would be helpful if you could provide relevant references.

Atherosclerosis is a major underlying cause of cardiovascular related death and

understanding how to ameliorate or prevent adverse complications from atherosclerosis, including plaque rupture or erosion, is a major focus of preclinical and clinical research. Historically, much of this focus has been on reducing cholesterol, leading to the widespread use of statins, and on detrimental inflammation. Recently, smooth muscle cells have become a target to increase plaque stabilization, and in the process, metabolic state and regulation has emerged as a major player in smooth muscle cell phenotypic switching and ECM deposition. The authors expand on previous work detailing the importance of metabolic state in atherosclerosis, and in SMC in particular. In this manuscript, the authors provide evidence that SMC to pDC phenotypic switching is regulated by KLF4, PFKFB3, eEF1A2, and circCTFP1.

While the authors clearly layout the role KLF4 plays in regulating SMC metabolic state in vitro, this was first shown, although not in as much detail, in previous work by Alencar et al (Circ 2020) where knockout of KLF4 in SMC showed significant changes in metabolic pathways by bulk RNAseq, as well as resulting in SMC modulation based on immunofluorescence and scRNAseq analyses. Other relevant papers in the field of atherosclerosis, metabolism, and SMC include: Newman et al Nature Metab 2021 (SMC phenotypic transitions and ECM synthesis are regulated by metabolic state), Tomas et al EHJ 2018 (metabolism in symptomatic vs non-symptomatic lesion), and the multiple papers from the 1980s including Lynch and Paul *Experientia* 1985 (energy metabolism and contractility ability in SMC). The novelty of this manuscript stems mostly from the identification of the small non coding RNA effectors of phenotypic switching, especially as it relates to post-RNA or post-translational modification of protein as a way to induce phenotypic changes via modulating metabolic state. This is quite exciting and this idea in particular will be important for the eventual goal of targeting individual cell types in disease.

However, the conclusions about phenotypic state as well as impact on atherosclerosis in this paper are not entirely supported by the data. Nevertheless, identification of this RNA modifier axis has the potential to influence thinking with substantial clarifications and/or revisions.

Is the work convincing, and if not, what further evidence would be required to strengthen the conclusions?

The authors outline putative SMC to pDC phenotypic switching and correlate that to atherosclerotic lesion progression. The SMC to pDC switch itself is, to this reviewer's knowledge, unknown previously. However, the identification of SMC-derived pDCs is solely reliant on marker protein co-localization of SMC marker proteins and pDC marker

proteins. This is problematic for a number of reasons including the ubiquity of many of the markers used by the authors in atherosclerotic lesions, the propensity of SMC themselves to downregulate their own marker proteins (rendering use of ACTA2 in vivo unreliable), and the relative inabundance of pDCs in atherosclerotic lesions (as cited in ref 32). Since this is a main point the authors are trying to make, it is imperative that evidence for this be incontrovertible.

Major comments:

Q1: SMC to pDC phenotypic switching. To this reviewer, the almost complete co-localization of ACTA2 staining with CD123 (pDC marker) is suspect as authors seem to imply that all ACTA2+ (putative SMC) have transitioned to a pDC phenotype. Couple this with the multiple SMC scRNAseq studies over the past few years (Wirka et al, Pan et al, Alencar et al) that do not show a pDC-like cluster derived from murine sorted lineage tagged SMC, this reviewer is not convinced of the author's claims of SMC to pDC transition in atherosclerosis. Further, there is no attempt to positively identify the ACTA2+ cells as SMC. This reviewer suggests confirming SMC origin by more than ACTA2 expression as well as providing more substantial evidence than marker staining that pDC-like cells exist, as this is the crux of the author's story. While the reviewer is aware that lineage tracing is impossible in humans, claims of SMC-derived pDC can be supported by either In Situ Hybridization with Proximity Ligation Assay (Gomez et al Nat Meth 2012) and/or use of the many murine SMC-lineage tracing models available.

A1: As the reviewer suggested, we performed the In Situ Hybridization (ISH) with Proximity Ligation Assay (PLA) for detecting H3K4dime of the *MYH11* promoter which is a highly specific marker of SMC lineage. Results showed that nearly 40% of cells within the lesions were PLA⁺ cells and nearly 30% of PLA⁺ cells were CD123⁺ cells (Supplementary Fig. 2a, b). This indicated that SMC-derived pDCs exist within the atherosclerotic plaques.

Because producing abundant interferon alpha (IFN- α) upon stimulation by foreign nucleic acids is a unique capacity of pDCs, we tested whether KLF4-overexpressing cells treated with CpG oligodeoxynucleotides (ODN) were able to produce IFN- α in the previous manuscript. Results showed that IFN- α -producing activities were hardly detectable in the KLF4-overexpressing VSMCs, and thus they remain distinguishable from the authentic pDCs. Consistently, a subset of studies showed that SMC-derived cells within atherosclerotic lesions express multiple markers of other cells, such as macrophages (M ϕ s) and mesenchymal stem cells (MSCs), but they did not appear to function as authentic M ϕ s or MSCs (Owens et al, Nat Med, 21(6):628-37; Fisher et al, Arterioscler Thromb Vasc Biol, 35(3):535-46). What the function of the phenotypically modulated VSMCs has and how their functional properties affect atherogenesis are of great challenge and need further studies in the future.

Q2: pDC specificity marker CD123. The idea of SMC can differentiate to a pDC-like cell is bolstered by the in vitro work showing increased CD123 and other pan DC mRNA expression (sometimes 3000-fold!) after KLF4 OE, however, authors do not evaluate any other marker of pDC (except for pan CD11c in mice) in vivo nor functionality aside from increased IFN α expression. Considering CD123 in disease is also used as a marker for hematopoietic stem cells, the reviewer suggests including additional metrics to determine a pDC-like state in vivo. Authors may consider including data using KLF4 KD vs. wild type in order to understand if this occurs normally, or is a function of the OE.

A2: Thank you for this good suggestion. Ly6D is recently recognized as a marker known to be expressed on pDCs (Musumeci et al, *Front Immunol*, 2019,10:1222; Rodrigues et al, *Mol Immunol*, 2020,126:25-30). In addition, Ly6D was largely induced by KLF4 overexpression in our study (Fig. 1e, f). Thus, we evaluated Ly6D expression in human atherosclerosis lesion and observed a co-staining of the *MYH11* promoter H3K4dime PLA and Ly6D (Supplementary Fig. 2c, d).

We had performed the loss-of-function experiments with siRNA vs KLF4, however, knockdown of KLF4 showed little effect on CD123 expression. A possible reason is that KLF4 overexpression might reach supra-physiological levels. However, the dose of adenovirus had been titrated and tracked relative to endogenous KLF4 levels before using. We reasoned that CD123 expression in control VSMCs was extremely low (Fig. 1) and could hardly be detected. Thus, its expression was easily induced but could not be reduced experimentally. In addition, the specificity of KLF4-induced pDC marker expressions was confirmed by the fact that KLF5 overexpression did not affect these markers (Fig. 1f).

Q3: IF staining controls. The reviewer questions the settings used for analysis of co-localization in Figures 1 and 8 and Supp Figs 1 and 5. The images presented look over-processed and/or over-blown. This especially apparent in the media and the body of the lesion where there is an unexpected morphology of the ACTA2 staining as well as the seemingly 100% co-localization of multiple markers including CD68, CD123, and ACTA2. Most previous reports, including ones the author cites (reference 4) show that about 10% of the lesion cells are ACTA2+ and that only a very small fraction of ACTA2+ cells co-stain for macrophage markers. The reviewer urges the authors to show the unprocessed images and to provide images of negative controls and more detailed information about assessment of co-localization and relative abundance of cell types in the methods.

A3: According to the reviewer 1's comment to show the co-staining of KLF4-CD123/CD68-ACTA2 in atherosclerosis lesion, we improved the experimental method and performed a multiplex tyramide signal amplification (TSA) staining, which is open and flexible for use with any primary antibody without cross-reactivity. We exhibited the unprocessed images in Figs. 1c, d and Supplementary Fig. 1. The staining data have

been quantified and statistically analyzed accordingly. As a result, we observed 19% and 12% of SM α -actin positive (SMA- α +) cells that expressed CD123 and CD68, respectively, in human atherosclerotic lesions. Also, we performed the In Situ Hybridization (ISH) with Proximity Ligation Assay (PLA) for detecting H3K4dime of the MYH11 promoter which is a highly specific marker of SMC lineage. Results showed that nearly 40% of cells within the lesions were PLA+ cells and nearly 30% of PLA+ cells were CD123+ cells. Experimental method and result were described, respectively, in Method and Result section.

Also, according to the reviewer 1's comment to present data from mouse tissue obtained close to the aortic valve (data was from the carotid artery of mouse in the previous manuscript), we performed immunofluorescence staining using mouse aortic root paraffin sections and the unprocessed images were shown in Figs. 8 and Supplementary Fig. 12d-f. Furthermore, a statistical quantification was also enclosed together with the representative images.

Q4: Quantification of claims:

a. Please quantify the percentage of ACTA2 cells that co-localize with the various markers in the figures (e.g. what is one to understand from "...obvious reduction in the expression of these genes"? page 9 line 345). Please also quantify plaque metrics to bolster claims of "enhance[d] atherosclerosis".

A4a: The immunofluorescence data and oil red O staining have been quantified and statistically analyzed in the revised version (Figs. 1d, 8, and Supplementary Fig. 1, 2, and 12). As a result, we observed 19% and 12% of SM α -actin positive (SMA- α +) cells that expressed CD123 and CD68, respectively, in human atherosclerotic lesions. In Situ Hybridization (ISH) with Proximity Ligation Assay (PLA) for detecting H3K4dime of the MYH11 promoter, a highly specific marker of SMC lineage, showed that nearly 40% of cells within the lesions were PLA+ cells and nearly 30% of PLA+ cells were CD123+ cells.

b. The reviewer suggests including a description of and relative abundance of pDC in plaques. The authors suggest that an increase in ACTA2+ CD123+ cells in atherosclerotic lesions in mice is associated with increased plaque size (without quantification), but aside from this, there is no clear stated directionality on if these metabolic changes are beneficial or detrimental, nor if changes (purportedly increases) in pDC-like cells are protective or pro-atherogenic (excluding plaque size, which is not the best indicator of stability, for references see the works of Virmani et al from CVPath).

A4b: The description of and relative abundance of pDCs in plaques were added in the first paragraph of the Result section in the revised version (**page 4 line 114-122**), showing that 19% of SM α -actin positive (SMA- α +) cells expressed CD123 in human atherosclerotic lesions. In Situ Hybridization (ISH) with Proximity Ligation Assay (PLA) for detecting H3K4dime of the MYH11 promoter, a highly specific marker of SMC lineage, showed that

nearly 40% of cells within the lesions were PLA⁺ cells and nearly 30% of PLA⁺ cells were CD123⁺ cells.

Q5: Conclusions. Because of the lack of lineage tracing, quantification of lesion area, cell phenotype, and negative controls for immunofluorescence images, the authors are suggested to tone down a number of the major conclusions in the paper.

a. On Page 10 paragraph 1, the authors suggest that ACTA2 staining is indicative of a SMC origin, but do not take into account the fact that in mice, nearly 80% of the SMC-derived cells have lost their ACTA2⁺ marker expression, especially those in the lesion core (ref 4).

A5a: As the reviewer suggested, we performed the ISH-PLA for detecting H3K4dime of the *MYH11* promoter which is a highly specific marker of SMC lineage. Results showed that nearly 40% of cells within lesions were PLA⁺ cells and nearly 30% of PLA⁺ cells were CD123⁺ cells (Supplementary Fig. 2a, b). Accordingly, the conclusion was revised as “Consistently, we here also observed CD123 expression in human atherosclerotic plaques and a co-localization of CD123 with VSMCs (page 10 line 367)”

b. In SFig 1, authors note that “pDC marker CD123 and macrophage marker CD68 are abundantly expressed in VSMCs...” indicating that CD123 and CD68 are co-expressed, but the idea that the pDC are derived from monocytes/macrophages is not explored. Indeed, monocytes and pDC share a known common progenitor (HSC, MPP), whereas SMC and pDC do not share a known progenitor, is it possible that 1. The pDC in the lesion are derived from the CD68 cells (i.e. myeloid cells that may have acquired ACTA2 expression, Newman Nat Metab 2021, Albarran-Juarez Atherosclerosis 2016, Caplice PNAS 2003) or 2. the co-localization of these three markers in the plaque is an artifact of staining?

A5b: Thank you very much for this point. We deleted this sentence in the revised version, because whether CD123 and CD68 are co-expressed in SMC-derived pDCs or in SMC-derived Mφs needs further studies in the future (also please see the response to c).

c. The role of KLF4 in SMC phenotypic switching is well-known, however the authors suggest that KLF4 exclusively regulates the SMC to pDC switch. Indeed, Alencar et al (Circ 2020) show a number of KLF4-regulated SMC-derived phenotypes in atherosclerosis including a chondrocyte-like cell and other osteogenic phenotypes but not a pDC-like phenotype. Nor is this shown in the other related scRNAseq studies of SMC-labeled cells in atherosclerosis (Wirka et al, Pan et al). Conclusions would be supported if authors show this unique KLF4-PFKFB3-RNA axis is specific for pDC transitions.

A5c: As we are known, in the past, the pDC markers were rarely detected within the atherosclerotic lesions except Daissormont's study showing that CD123 stained cells were found to show considerable co-localization with VSMC (Daissormont et al, Circ Res,

2011, 109:1387-1395) and Owen's study showing that VSMCs can express dendritic cell marker ITGAX (CD11c) (Owens et al, Nat Med, 21(6):628-37). It is not excluded that the VSMCs-derived plaque cells may share the common markers, including the well-known Mφs marker, chondrocyte marker and the less-known DC marker, which needs to be addressed in future studies. Whether the KLF4-PFKFB3-RNA axis is specific for pDC transitions will be investigated in future work.

Minor comments:

Q1: Please define PFKFB3 page 4 line 107

A1: We have introduced PFKFB3 in the Introduction section in the revised version (**page 3 line 93**).

Q2: Figure 1c, SFig 1a, SFig 5: please indicate if the field of view is in the core or the cap region of the lesion

A2: The field of view is in the core region of the lesion, and this has been described in the corresponding figure legends.

Q3: Please add graphs quantifying the overlap of ACTA2 cells that co-localize with the various markers in the figures (i.e. what is one to understand from "...obvious reduction in the expression of these genes"? page 9 line 345)

A3: The statistical quantifications (bar graphs) have been enclosed together with the representative images (Supplementary Fig. 12d-f).

Q4: Please add detail to the methods about cell culture conditions including duration of treatments

A4: The detail of cell culture conditions including duration of treatments have been added to the Cell culture and treatment part in Methods section (**page 12 line 479**).

Q5: Please ensure conclusions are borne out in the data or clarify/tone down conclusions not directly show in the manuscript. For example: clarify claim on pg 11 line 429 that the "...glycolytic shift...can also activate Akt" by showing these metabolic using Seahorse with glycolysis agonists and KLF4 OE (refer to page 8 line 286, where it "evidently" causes changes).

A5: We have rewritten this sentence as "Seahorse glycolysis stress tests were performed and we observed that KLF4 overexpression could increase the extracellular acidification rate (ECAR) related to both glycolysis and glycolytic capacity (Fig. 2e, f)".

Q6: Regarding Akt expression in atherosclerosis pathogenesis, authors show Akt is phosphorylated after KLF4 OE as part of the PFKFB3 axis and later suggest that this accelerates atherosclerosis pathology. A seminal paper in the field (Fernandez-Hernando ATVB 2009) shows Akt1 is required for SMC survival and for overall plaque stabilization. How do the authors claims fit in with the wider field?

A6: In this study, we show that glycolytic shift driven by KLF4 and PFKFB3 facilitates the

phenotypic switching of VSMCs via activating Akt phosphorylation. Our findings, i.e., that glycolytic shift facilitates the phenotypic switching of VSMCs via activating Akt signaling, are not contradictory to the previously reported results showing that Akt1 is required for SMC survival and for overall plaque stabilization (Fernandez-Hernando ATVB 2009). It is well known that the phenotype switching of VSMCs and proliferation are the prerequisites of vascular remodeling and related CVDs, including atherosclerosis, hypertension, restenosis, etc. VSMC survival in the fibrous cap of the atherosclerotic plaque is beneficial for fibrous cap formation and plaque stability, whereas abnormal proliferation and accumulation of VSMCs in the plaque core accelerates atherosclerosis pathology because VSMCs in the plaque core transform into macrophage-like VSMCs and foam cells, and these cells are detrimental for plaque stability (Mandy O J Grootaert , Martin R Bennett , *Cardiovasc Res.* 2021 Sep 28;117(11):2326-2339).

Q7: The authors should include in the figure legends statistical tests performed, error bars (SEM or SD?), replicate numbers, and scale bars for all images to facilitate assessment of statistical analysis and validity of data.

A7: According to your suggestion, the corresponding descriptions have been added to the figure legends.

Reviewers' comments:

Reviewer #1 (Remarks to the Author):

The authors have satisfied most of my previous concerns. However, I still have two minor points:

1- I am not very convinced of the in situ/IHC staining shown in figure 5J. The circRNA signal looks very faint, as it was not specific.

2- The authors should use the comment to the response to my Q12 to improve the flow of the text. I would add the reference they have included in the point-to-point letter and I would tone down the sentence (page 8, line 280): "This might imply that the effect...."

Reviewer #2 (Remarks to the Author):

All questions are being answered and all comments were sufficiently addressed, while it is a pity that reduction in KLF4 is not able to reduce CD123 expression.

Reviewer #3 (Remarks to the Author):

Specific Comments for current manuscript

The authors made substantial additions to their original paper, addressing almost all comments from the three reviewers. The idea, novelty, and implications for the field are still interesting and relevant (including identifying ways to more specifically target transcription factor co-factors, as TF are not good candidates for therapy), however a number of points, especially related to the atherosclerotic lesion must be addressed. They are as follows:

Major points:

1. The figures need to be formatted more logically. This includes significant layout changes and clear demarcation of panels that go together (i.e. WB + graph of densitometry), putting panels in order rather than jumping around the page (i.e. go left to right or up to down, but not up then down, then up and right, then down, etc). An example of a particularly difficult figure to parse out is 3, where (e) comes after (f), and 4, where (e) comes before (b), (c), and (d).

2. One large conclusion in this paper is that there is a relatively high percentage of smooth muscle-derived cells are pDC-like, however there are two immediate problems with this conclusion. First, according to the manufacturer, use of the TSA kit precludes making quantitative assessments of cell populations because of the fact that the signal is amplified in order to be detectable. Second, the ISH-PLA assay is not 100% efficient and thus, any semi-quantitative conclusions made MUST include an assessment of the efficiency of the assay by assessing the % PLA + / DAPI+ cells in the media (see original work by Gomez, Shankman, et al, Nat Meth, 2012). Thus, the claim that 30% of the H3K4me2 PLA+ cells in the lesion are CD123+ should be reassessed, or if this percentage is a representation post-correction, the correction equation and efficiency must be included in the methods, text, and/or legend. If these conditions cannot be satisfied, the authors should temper their conclusions. It does seem as if some ACTA2+ and/or SMC-derived cells do indeed co-localize with pDC markers, but the authors must substantiate claims of these very high relative percentages of cells (for reference, only 20% of the SMC in the mouse lesion are SMC-like, expressing ACTA2 per Shankman et al, Nat Med, 2015).

3. Significant detail is missing from the methodology (and outlined specifically below) to facilitate the ability of a researcher to reproduce the work.

Minor points:

1. Please indicate if n=3 means a biological or technical replicate

2. Please make scale bars thicker and unobscured by area of interest boxes
3. Images of lesions should include dotted outlines separating the media, core, and lumen. It is very difficult to determine what many of the panels represent, thus the reviewer can often not come to the stated conclusions about the data.
4. Please be consistent in representing control and experimental groups, sometimes the control is on the left, sometimes it is on the right (example: figure 3a and b). Similarly, be consistent with labeling of graphs and legends as with the coloring used to represent the staining (sometimes ACTA2 is purple, sometimes red).
5. In figure 5, please actually indicate the constructs. It is not clear from the text or the legend what the other constructs are besides #1 and #2, which are indeed indicated in the text.
6. All immunofluorescent images need to be depicted with proper contrast. Dark purple for ACTA2 is not visible, consider changing to grey or cyan, again keep pseudocoloring consistent throughout panels and figures.
7. Similarly, in figure 5j, the red staining is not easily visible, and showing the overlay does not readily show overlap between eEF1A2 and circCTDP1. Adding arrows would help. Same scale bar issue here.
8. Please indicate the dosage used for time course experiments instead of saying "indicated dosages". Example: Figure 7b.
9. The images of atherosclerosis in Figure 8 and Supplemental Figures 1, 2, and 12 have the following issues and/or do not accurately represent the data in the graphs/text in the following ways:
 - a. Most of the mouse images do not look like the aortic root. Therefore, authors must include the whole root and/or lesion and not just one field of view or correct the region. Furthermore, the authors sometimes talk about the "bifurcations"
 - b. Panel 8a, Apoe^{-/-} + Choline diet is showing a region in the inner layer of the media. Furthermore, it seems as if all cells are CD11c⁺. Inset is missing scale bar.
 - c. Supplemental figure 1: inset is missing a scale bar. Are the percentages in (b) really less than 1%, or do they represent ~80%, ~20%, ~20%, respectively? Please change the purple grey or cyan.
 - d. Supplemental figure 2: same comment about purple, arrows, and better contrast would help. Figure 2a also must have a correction factor for PLA, which is NOT 100% efficient (closer to 60%). This can go in the methods section, but it is missing from the methods section currently.
 - e. Supplemental figure 12: all panels are not representative of the data, seem to show 100% staining of the green marker, and nearly 0% staining of ACTA2.
 - i. panel d, row 4: this reviewer cannot tell if the panel is showing the inner layer of the media or the lesion core. These panels do not show any ACTA2 at all (even in the inset), and therefore are not representative of the graph. Inset is missing a scale bar.
 - ii. Panel e, row 4: this inset is most definitely the inner layer of the media. Also, there is no ACTA2 staining in the representative image, barely any cells, and therefore not representative of the data in the graph. Inset is missing a scale bar.
 - iii. Panel f, row 4: this inset is indeed the core, but again, completely lacking in ACTA2⁺ cells. Inset is missing a scale bar.
10. The abstract is confusing as written, please consider rewording.
11. The methods seem to have major editing errors, especially in the ISH-PLA section, please check over carefully.
12. Authors say that CD123 and KLF4 are abundantly expressed, but show a % population of 0.8, 0.2, and 0.2 in Figure 1. Is this right?
13. Please indicate if glucose and other metabolites were in the media for glycolytic/lactate assays. This is not apparent in the methods section and important when assessing metabolic state and changes in cells.
14. Minor grammatical and editing issues in manuscript.

Specific comments for major comments in author's rebuttal

- Q1. Authors did perform suggested IF and quantification studies, but limitations are outlined in major comments point 2.

Q2. Authors addressed comments.

Q3. It is more important to include negative controls not just unprocessed images when using a TSA kit as it is not recommended for quantification per the manufacturer. Furthermore, the unprocessed images seem to show 100% or near 100% staining of all the markers and nearly 0% staining of ACTA2 in multiple panels. This makes it difficult to critically assess claims made in the text and/or graphs quantifying % populations.

Q4. No additional comments other than previously stated caveats.

Q5. Authors addressed all comments.

Point-by-point response to reviewer 1

Reviewer #1 (Remarks to the Author):

The authors have satisfied most of my previous concerns. However, I still have two minor points:

Q1: I am not very convinced of the in situ/IHC staining shown in figure 5J. The circRNA signal looks very faint, as it was not specific.

A1: We thank the reviewer's comment. Under the same parameter settings for fluorescence microscope, the red fluorescence intensity (circCTDP1) is weaker than the green fluorescence (eEF1A2), probably because circCTDP1 was relatively lowly expressed. However, we did observe the red fluorescence spots scattered along nuclear envelope, albeit faint. In particular, when green fluorescence of eEF1A2 and red fluorescence of circCTDP1 were merged as a yellow color and were locally enlarged, we indeed observed a clear co-localization between eEF1A2 and circCTDP1, as indicated by the white arrow (Fig. 5j) in the revised version, implying that circCTDP1 is expressed in HVSMCs.

Q2: The authors should use the comment to the response to my Q12 to improve the flow of the text. I would add the reference they have included in the point-to-point letter and I would tone down the sentence (page 8, line 280): "This might imply that the effect....."

A2: Thank you for your good suggestion. The reference had been added in the revised manuscript (reference 29). As the reviewer suggested, we toned down the conclusion as "This might imply that the effect....." (page 8 line 280).

Point-by-point response to reviewer 2

Reviewer #2 (Remarks to the Author):

All questions are being answered and all comments were sufficiently addressed, while it is a pity that reduction in KLF4 is not able to reduce CD123 expression.

A: Thank you very much for taking your time to review our manuscript.

Point-by-point response to reviewer 3

Reviewer #3 (Remarks to the Author):

Specific Comments for current manuscript

The authors made substantial additions to their original paper, addressing almost all comments from the three reviewers. The idea, novelty, and implications for the field are still interesting and relevant (including identifying ways to more specifically target transcription factor co-factors, as TF are not good candidates for therapy), however a number of points, especially related to the atherosclerotic lesion must be addressed. They are as follows:

Major points:

Q1: The figures need to be formatted more logically. This includes significant layout changes and clear demarcation of panels that go together (i.e. WB + graph of densitometry), putting panels in order rather than jumping around the page (i.e. go left to right or up to down, but not up then down, then up and right, then down, etc). An example

of a particularly difficult figure to parse out is 3, where (e) comes after (f), and 4, where (e) comes before (b), (c), and (d).

A1: As the reviewer suggested, the figures have been formatted more logically. The panels were rearranged from left to right and up to down, and the panels of WB + graph of densitometry were demarcated more clearly (Please refer to Figs. 4-7 and Supplementary Figs. 4, 8, 10 and 11).

Q2: One large conclusion in this paper is that there is a relatively high percentage of smooth muscle-derived cells are pDC-like, however there are two immediate problems with this conclusion. First, according to the manufacturer, use of the TSA kit precludes making quantitative assessments of cell populations because of the fact that the signal is amplified in order to be detectable. Second, the ISH-PLA assay is not 100% efficient and thus, any semi-quantitative conclusions made MUST include an assessment of the efficiency of the assay by assessing the % PLA + / DAPI+ cells in the media (see original work by Gomez, Shankman, et al, Nat Meth, 2012). Thus, the claim that 30% of the H3K4me2 PLA+ cells in the lesion are CD123+ should be reassessed, or if this percentage is a representation post-correction, the correction equation and efficiency must be included in the methods, text, and/or legend. If these conditions cannot be satisfied, the authors should temper their conclusions. It does seem as if some ACTA2+ and/or SMC-derived cells do indeed co-localize with pDC markers, but the authors must substantiate claims of these very high relative percentages of cells (for reference, only 20% of the SMC in the mouse lesion are SMC-like, expressing ACTA2 per Shankman et al, Nat Med, 2015).

A2: We thank the reviewer's comment. The multiplex TSA staining methods involve Tyramide signaling amplification with Opal reactive fluorophores. In this process, the fluorescence intensity is specifically and highly amplified, but the number of positive cells is hardly affected. A number of references indeed make quantifications of this multiplexed IHC staining to quantify cell density (Carvelli J et al, Nature, 2020;588(7836):146-150; Ying L et al, Oncoimmunology, 2018;7(6):e1433520).

In this manuscript, the quantification of ISH-PLA assay is not a representation of post-corrected results. Indeed, as the reviewer's comment, semi-quantitative conclusions made must include an assessment of the efficiency of the assay by assessing the % PLA+ /DAPI+ cells in the media. Unfortunately, we just captured images and quantified the positive cells in atherosclerotic lesions but not in the media of the human tissues, when performing experiments. Considering the fact that this study indeed demonstrated a co-localization of SMC marker and pDC marker, we tempered our conclusion and revised it as "As a result, we observed a PLA signal in CD123+ and LY6D+ (another pDC marker)

cells" (page 4 line 119). Simultaneously, we marked the quantification of ISH-PLA assay as an un-corrected result in the legends of Supplementary Fig. 2b and d.

Q3: Significant detail is missing from the methodology (and outlined specifically below) to facilitate the ability of a researcher to reproduce the work.

A3: To facilitate the ability of a researcher to reproduce the work, all the concerns (outlined specifically below) suggested by the reviewer have been addressed.

Minor points:

Q1: Please indicate if n=3 means a biological or technical replicate.

A1: n=3 means a biological replicate.

Q2: Please make scale bars thicker and unobscured by area of interest boxes.

A2: Thicker and unobscured scale bars have been marked in area of interest boxes.

Q3: Images of lesions should include dotted outlines separating the media, core, and lumen. It is very difficult to determine what many of the panels represent, thus the reviewer can often not come to the stated conclusions about the data.

A3: Images of human vascular tissues showed the atherosclerotic lesion region (Figs. 1c, 3l and Supplementary Fig. 1, 2). Images of mouse arterial lesions have been outlined by dotted lines to separate the media, core, and lumen (row 4 of Fig. 8a and Supplementary Fig. 12d, e, f).

Q4: Please be consistent in representing control and experimental groups, sometimes the control is on the left, sometimes it is on the right (example: figure 3a and b). Similarly, be consistent with labeling of graphs and legends as with the coloring used to represent the staining (sometimes ACTA2 is purple, sometimes red).

A4: We thank the reviewer's suggestion. We have adjusted the position of control and experimental groups, and the control group is shown on the left and experimental group on the right (Figs. 1e, 3a and b).

According to this suggestion raised by the reviewer and considering Q6 below, to enhance the contrast of immunofluorescent images, we changed the color of ACTA2 to grey in Figs. 1c, 3l and Supplementary Figs. 1, 2. Because images of mouse arterial lesion were only involved in three colors including DAPI, we did not change the color of ACTA2 in these images to keep the red-green contrast (Fig. 8a and Supplementary Figs. 12d, e, f). Also, we changed the color of CD123 to red in Supplementary Fig. 2 to make it consistent with that in Fig. 1c. Accordingly, the labeling of graphs and legends have been made consistent with the coloring used to represent the staining.

Q5: In figure 5, please actually indicate the constructs. It is not clear from the text or the legend what the other constructs are besides #1 and #2, which are indeed indicated in the text.

A5: The constructs of circRNAs have been indicated in the legend of Fig. 5a.

Q6: All immunofluorescent images need to be depicted with proper contrast. Dark purple for ACTA2 is not visible, consider changing to grey or cyan, again keep pseudocoloring consistent throughout panels and figures.

A6: Thanks for your suggestion. Same as the answer to Q4. Please see A4.

Q7: Similarly, in figure 5j, the red staining is not easily visible, and showing the overlay does not readily show overlap between eEF1A2 and circCTDP1. Adding arrows would help. Same scale bar issue here.

A7: Under the same parameter settings for fluorescence microscope, the red fluorescence intensity (circCTDP1) was weaker than the green fluorescence (eEF1A2), probably because circCTDP1 was relatively lowly expressed. However, when green fluorescence of eEF1A2 and red fluorescence of circCTDP1 were merged as a yellow color and were locally enlarged, we indeed observed a clear co-localization between

eEF1A2 and circCTDP1. We added the white arrows to show the overlap between eEF1A2 and circCTDP1 (Fig. 5j) in the revised version. Additionally, Thicker and unobscured scale bars have been marked.

Q8: Please indicate the dosage used for time course experiments instead of saying “indicated dosages”. Example: Figure 7b.

A8: 200 mM TMAO was used for the time course experiments and we have indicated this dosage in Fig. 7b.

Q9: The images of atherosclerosis in Figure 8 and Supplemental Figures 1, 2, and 12 have the following issues and/or do not accurately represent the data in the graphs/text in the following ways:

a. Most of the mouse images do not look like the aortic root. Therefore, authors must include the whole root and/or lesion and not just one field of view or correct the region. Furthermore, the authors sometimes talk about the “bifurcations”

A9a: Supplementary Fig. 12b shows a representative en face analysis of atherosclerotic lesions in the whole aorta stained with Oil Red O and Supplementary Fig. 12c shows the cross sectional analysis of atherosclerotic lesions in the carotid arteries. All the mouse ascending aorta were embedded in paraffin, thus, it is difficult to perform Oil red O staining to observe the whole lesion. Slices obtained from the proximal end of paraffin-embedded ascending aorta close to the aortic valve were used for immunofluorescent staining. Thus, all the mouse immunofluorescent staining images are the aortic root. However, we just showed one field of view to locally observe the protein expression in the lesion without imaging the whole section (Fig. 8a and Supplementary Fig. 12d, e, f).

b. Panel 8a, Apoe^{-/-} + Choline diet is showing a region in the inner layer of the media. Furthermore, it seems as if all cells are CD11c⁺. Inset is missing scale bar.

A9b: This image has been outlined by dotted lines to separate the media, core, and lumen (row 4 of Fig. 8a). We considered the region between the two white dotted lines as the lesion. Indeed, all cells in the interest box are CD11c⁺, however, most of the cells, especially those adjacent to the lumen in the lesion region, are not CD11c⁺. Scale bar has been added to the inset.

c. Supplemental figure 1: inset is missing a scale bar. Are the percentages in (b) really less than 1%, or do they represent ~80%, ~20%, ~20%, respectively? Please change the purple grey or cyan.

A9c: Scale bar has been added to the inset. The percentages in Supplementary Fig. 1b represent ~80%, ~20%, ~20%, respectively. We made a mistake in drawing the chart. Also, the same problem occurred in Fig. 1d. Scale of the ordinate of these two figures has been corrected. The purple has been changed to grey.

d. Supplemental figure 2: same comment about purple, arrows, and better contrast would help. Figure 2a also must have a correction factor for PLA, which is NOT 100% efficient (closer to 60%). This can go in the methods section, but it is missing from the methods section currently.

A9d: The purple has been changed to grey. White arrows were added to the inset to show the PLA⁺CD123⁺ and PLA⁺LY6D⁺ cells.

About the efficiency and quantification of the ISH-PLA assay, same as the answer to Q2. Please see A2.

e. Supplemental figure 12: all panels are not representative of the data, seem to show 100% staining of the green marker, and nearly 0% staining of ACTA2.

i. panel d, row 4: this reviewer cannot tell if the panel is showing the inner layer of the media or the lesion core. These panels do not show any ACTA2 at all (even in the inset), and therefore are not representative of the graph. Inset is missing a scale bar.

A9ei: The images have been outlined by dotted lines to separate the media, core, and lumen (row 4 of Supplemental Fig.12d,e,f). Just as the reviewer raised, these panels mainly showed the media. In fact, we observed that KLF4 was abundantly expressed in the media but not in the lesion core in this mouse model. We reasoned that high expression of KLF4 might be an early event that drives VSMC proliferation, migration and phenotypic switch. The description in the text was inaccurate in the previous version of the manuscript, and we revised it as “In addition, we also observed that 16% of cells in the media were KLF4+SMA- α ⁺ and 25% and 21% of cells in the atherosclerotic lesions were eEF1A2+SMA- α ⁺ and PFKFB3+SMA- α ⁺, respectively” (page 9 line 347).

We also observed that the expression of ACTA2 was extremely low in the inner layer of the media with high expression of KLF4. This was not caused by nonspecific staining, because we could detect a certain expression of ACTA2 in the media. We speculated that this might be because KLF4 was highly expressed in the media and it could inhibit ACTA2 expression. Also, it might be because the expression of ACTA2 was decreased with the progress of the disease and its expression was extremely low in this stage of this model. Further study to investigate the relationship between time-course changes of KLF4 and ACTA2 expression and the disease severity may be useful to help to solve this problem.

Scale bar has been added to the inset.

ii. Panel e, row 4: this inset is most definitely the inner layer of the media. Also, there is no ACTA2 staining in the representative image, barely any cells, and therefore not representative of the data in the graph. Inset is missing a scale bar.

A9eii: This image has been outlined by dotted lines to separate the media, core, and lumen. A new inset from the lesion core has been added. As we mentioned in A9ei, the expression of ACTA2 was extremely low and the immunofluorescent signal was very faint. However, when red fluorescence of ACTA2 and green fluorescence of eEF1A2 were merged as a yellow color and were locally enlarged, we indeed observed a clear co-localization between ACTA2 and eEF1A2, implying that there is low expression of ACTA2 in the lesion core, albeit at low cell numbers. White arrows were added to indicate the ACTA2 staining.

Scale bar has been added to the inset.

iii. Panel f, row 4: this inset is indeed the core, but again, completely lacking in ACTA2+ cells. Inset is missing a scale bar.

A9eiii: This image has been outlined by dotted lines to separate the media, core, and lumen. As we mentioned in A9ei, the expression of ACTA2 was extremely low and the immunofluorescent signal was very faint. However, when red fluorescence of ACTA2 and green fluorescence of PFKFB3 were merged as a yellow color and were locally enlarged, we indeed observed a co-localization between ACTA2 and PFKFB3, implying that there is low expression of ACTA2 in the lesion core, albeit at low cell numbers. White arrow was added to indicate the ACTA2 staining.

Scale bar has been added to the inset.

Q10: The abstract is confusing as written, please consider rewording.

A10: We deleted the sentence “Genetic depletion of PFKFB3 blocks glycolytic shift driven by KLF4” in the abstract of the revised manuscript to make the abstract more concise (page 2 line 43).

Q11: The methods seem to have major editing errors, especially in the ISH-PLA section, please check over carefully.

A11: Thanks for your careful review. The ISH-PLA assay was incorrectly edited due to our negligence. We have checked over the methods carefully and have re-written this part (page 15 line 586).

Q12: Authors say that CD123 and KLF4 are abundantly expressed, but show a % population of 0.8, 0.2, and 0.2 in Figure 1. Is this right?

A12: We are very sorry for our incorrect ordinate scale. Correction has been made in the revised version of manuscript. Please see A9c.

Q13: Please indicate if glucose and other metabolites were in the media for glycolytic/lactate assays. This is not apparent in the methods section and important when assessing metabolic state and changes in cells.

A13: Glucose was in the media for lactate assay. ECAR was measured using the Seahorse XF glycolysis stress test Kit (Agilent) in XF base medium containing 10 mM glutamine without glucose. We added the relevant description in the methods section (page 17 line 680 and 685).

Q14: Minor grammatical and editing issues in manuscript.

A14: We have checked over the manuscript carefully.

Reviewers' comments:

Reviewer #1 (Remarks to the Author):

The authors have satisfied my concerns. I have no further requests.

Reviewer #3 (Remarks to the Author):

To reiterate, the idea, novelty, and implications for the field are interesting, important, and relevant. However, while the authors have revised the manuscript text per the latest comments, this reviewer still has significant issues with the immunofluorescence images; issues that have been expounded upon in the last two reviews. Briefly, a lot of the staining is over or under exposed and thus requires negative control images and most panels cannot be considered representative images. (Where is the lesion in the choline+DMB IF images, and thus, where are the double positive CD11c+ ACTA2+ cells?)

Reviewers' comments:

Reviewer #1 (Remarks to the Author):

The authors have satisfied my concerns. I have no further requests.

A: Thank you very much for taking your time to review our manuscript.

Reviewer #3 (Remarks to the Author):

To reiterate, the idea, novelty, and implications for the field are interesting, important, and relevant. However, while the authors have revised the manuscript text per the latest comments, this reviewer still has significant issues with the immunofluorescence images; issues that have been expounded upon in the last two reviews. Briefly, a lot of the staining is over or under exposed and thus requires negative control images and most panels cannot be considered representative images. (Where is the lesion in the choline+DMB IF images, and thus, where are the double positive CD11c+ ACTA2+ cells?)

A: Thanks for your careful review. We have replaced the choline+DMB IF images in Fig. 8a by a set of more representative images with a lesion. In addition, negative control images have been supplemented in the revised version (Supplementary Figs. 1c, d, 2c, and 12g, h).